# Online Social Welfare Function-based Resource Allocation

**Kanad Pardeshi** [1]    **Samsara Foubert** [2]    **Aarti Singh** [1]

## Abstract

In many real-world settings, a centralized decision-maker must repeatedly allocate finite resources to a population over multiple time steps. Individuals who receive a resource derive some stochastic utility; to characterize the population-level effects of an allocation, the expected individual utilities are then aggregated using a social welfare function (SWF). We formalize this setting and present a general confidence sequence framework for SWF-based online learning and inference, valid for any monotonic, concave, and Lipschitz-continuous SWF. Our key insight is that monotonicity alone suffices to lift confidence sequences from individual utilities to anytime-valid bounds on optimal welfare. Building on this foundation, we propose SWF-UCB, a SWF-agnostic online learning algorithm that achieves near-optimal $\tilde{\mathcal{O}}(n + \sqrt{nkT})$ regret (for $k$ resources distributed among $n$ individuals at each of $T$ time steps). We instantiate our framework on three normatively distinct SWF families: Weighted Power Mean, Kolm, and Gini, providing bespoke oracle algorithms for each. Experiments confirm $\sqrt{T}$ scaling and reveal rich interactions between $k$ and SWF parameters. This framework naturally supports inference applications such as sequential hypothesis testing, optimal stopping, and policy evaluation.

## 1. Introduction

Decision-makers routinely face the problem of allocating limited resources to a population over time: a park service assigning ranger patrols to backcountry zones, a chronic care program allocating community health worker visits among patients, or a school district allotting tutoring slots among students. Evaluating such allocations requires aggregating individual outcomes into a coherent measure of collective welfare. Social welfare functions (SWFs) provide a principled method for this aggregation, with axiomatic foundations from welfare economics that encode fairness-efficiency tradeoffs in a form amenable to optimization. Recent ML work has formalized SWFs within learning theory, providing sample complexity bounds for welfare estimation from batch data (Cousins, 2021; Pardeshi et al., 2024).

Real-world resource allocation is inherently sequential: individual utilities are unknown and must be learned from observed outcomes, yet decisions cannot be deferred while data accumulate. Each resource allocation affects the population's welfare, and losses from suboptimal decisions accumulate over time. There are two intertwined challenges. First, decision-makers must allocate resources effectively even when utility estimates are uncertain, somehow balancing strategies to reduce uncertainty with the real costs of suboptimal assignments. Second, they need valid statistical assessments and guarantees of welfare on demand: has the current policy achieved their desired welfare threshold? Is uncertainty low enough to commit to the current policy and deploy it at scale?

These questions arise adaptively, driven by externalities such as budget cycles, stakeholder reviews, or shifting priorities. Both challenges therefore demand statistical guarantees that remain valid regardless of when they are invoked. Recent work has begun studying online welfare optimization in bandits, aggregating welfare across the temporal sequence of decisions to measure fairness over time (see Section 2). We instead consider settings where a fixed population receives resources repeatedly, and welfare measures fairness within this persistent population at each decision point. This fixed-population formulation has distinct structure: allocation policies are continuous probability vectors over individuals rather than discrete arm selections, multiple resources are distributed per round, and decisions are coupled through the constraint that allocation probabilities must sum to the number of available resources. The optimal policy thus depends on the relative utilities across all individuals, creating interdependencies absent from standard bandit problems. We address both challenges through a unified confidence sequence-based framework.

Confidence sequences are interval estimates that remain

[1]Machine Learning Department, Carnegie Mellon University [2]Software and Societal Systems Department, Carnegie Mellon University. Correspondence to: Kanad Pardeshi <kpardesh@andrew.cmu.edu>.

*Proceedings of the 43$^{rd}$ International Conference on Machine Learning*, Seoul, South Korea. PMLR 306, 2026. Copyright 2026 by the author(s).

valid at arbitrary stopping times, a property useful for guiding policy decisions or certifying statistical conclusions. Our framework requires three natural assumptions on SWFs—monotonicity, concavity, and Lipschitz continuity—each playing a distinct, minimal role. Our key insight is that monotonicity of the SWF alone suffices to lift coordinate-wise confidence sequences for individual utilities into anytime-valid bounds on population welfare of the current allocation (Theorem 4.1). This lifting principle connects a versatile statistical primitive to information that decision-makers need: guarantees on the optimal welfare value over the expected utilities that hold uniformly over time. The same machinery that enables optimistic allocation policies for online learning also supports inference applications—sequential hypothesis testing, optimal stopping, and policy evaluation—as natural byproducts.

We make the following contributions. First, we formalize the problem of resource allocation over a population over multiple time steps, with the objective being given by an SWF-based aggregation of the ex-ante utilities. Second, we establish a confidence sequence lifting theorem that provides anytime-valid welfare bounds under monotonicity alone. Third, we instantiate this framework on three axiomatically distinct SWF families—Weighted Power Mean, Kolm, and Gini, spanning the spectrum from utilitarian to egalitarian objectives—and develop bespoke efficient oracles for policy optimization in each case. Fourth, we propose SWF-UCB, a general algorithm for online welfare maximization, and prove that it achieves near-optimal $\tilde{\mathcal{O}}(n + \sqrt{nkT})$ regret for $k$ resources over a population of $n$ in time $T$. Fifth, we present experiments confirming the predicted $\sqrt{T}$ scaling and revealing that intermediate values of $k$ yield the highest regret—a non-monotonic dependence suggesting rich structure in how resource scarcity interacts with learning difficulty. Finally, we describe how this confidence sequence framework can be applied to meaningful inference tasks such as sequential hypothesis testing (Section 7), optimal stopping (Appendix A.1.1, and policy evaluation (Appendix A.1.2).

## 2. Related Work

Our framework unifies several research threads spanning multi-armed bandits, fair allocation, and welfare economics, through a common lens of online welfare maximization and time-uniform statistical inference. However, unlike prior work on welfare-aware or fair bandits, which either focus on specific objectives or learning guarantees alone, our approach supports general monotone SWFs, provides near-optimal regret guarantees, and enables anytime-valid inference for optimized welfare under partial feedback.

**Multi-armed bandit variants.** Multi-play bandits (Anantharam et al., 1987; Gai et al., 2012) consider the sum of the arm rewards when multiple arms can be pulled, which corresponds to utilitarian welfare in our setting. Combinatorial bandits (Cesa-Bianchi & Lugosi, 2012; Kveton et al., 2015) study finding the best set of arms to pull for the greatest reward. To contrast, in our setting, all individuals need to be given a resource with non-zero probability, and thus we find the best *allocation* probabilities for the greatest welfare. (Chen et al., 2013) study combinatorial bandits with non-linear super-arm rewards assuming monotonicity and bounded smoothness. We use monotonicity of SWFs to establish a general confidence sequence lifting result (Theorem 4.1) and develop efficient and exact oracles for optimal allocation in our algorithm (SWF-UCB).

**Fair multi-armed bandits.** Fairness in multi-armed bandits usually involves each arm being played a minimum number of times (Chen et al., 2020; Wang et al., 2021) or meritocratic fairness (Joseph et al., 2016; 2018). (Lim et al., 2024) study multi-play multi-armed bandits, where each play corresponds to a user, and they measure regret using the performance of the worst-off user, i.e., egalitarian welfare. Our work subsumes egalitarian welfare and provides a framework for learning and inference when allocating resources among a population.

**Welfare-based regret in bandits.** (Barman et al., 2023) and (Sawarni et al., 2023) introduce Nash regret for the bandit setting, measuring cumulative loss via the geometric mean of *per-round regrets*. (Sarkar et al., 2025) and (Krishna et al., 2025) extend this idea to the $p$-mean welfare family. This line of work aggregates utilities across time steps, whereas our work aggregates utilities across the population at every time step. We also provide a general framework for different social welfare functions, and the $p$-mean welfare family arises as a special case of the welfare families we consider, under a different axis of aggregation.

**Social welfare functions.** The axiomatic foundations of our SWF families originate in welfare economics. (Atkinson, 1970) introduced inequality-averse welfare functions parameterized by a single parameter controlling the equity-efficiency tradeoff: our WPM family. (Kolm, 1976) characterized the Kolm-Pollak family via translation invariance, while (Weymark, 1981) formalized generalized Gini welfare through rank-dependent weights. In machine learning, (Cousins, 2021; 2023) provide Hölder continuity bounds enabling PAC-style learning guarantees. (Pardeshi et al., 2024) addressed the complementary problem of *learning* SWFs from preference data. To our knowledge, our work contributes the first online learning and time-uniform inference framework for allocation using these welfare families with provable regret guarantees.

# 3. Problem setup

We consider a population of $n$ individuals among which a centralized decision-maker allocates identical, indivisible resources at discrete time steps. At each time step $t$, $k \leq n$ resources arrive and are distributed among the population. Each individual receives at most one resource, and the utility of individual $i$ when receiving a resource at time $t$ is $u_{t,i} \overset{iid}{\sim} \mathcal{D}_i$, where $\mathcal{D}_i$ is an unknown distribution with mean $\mu_i = \mathbb{E}_{U \sim \mathcal{D}_i}[U]$. We assume that $\mu_i > 0$, and the utility of an individual not receiving a resource is zero.

Allocation occurs according to a (randomized) *policy*, given by the vector $\mathbf{p} \in [0,1]^n$, where $\sum_i p_i = k$. Each $p_i$ is the marginal probability that individual $i$ receives one of the $k$ resources. The *ex-ante* expected utility of the policy $\mathbf{p}$ for individual $i$ is thus given by $\mu_i p_i$. We denote the vector of expected utilities by $\boldsymbol{\mu} \odot \mathbf{p}$.

Our model assumes that the distributions $\{\mathcal{D}_i\}_{i \in [n]}$ of individual utilities remain static and do not change with time. Instead, we allow the allocation policy $\mathbf{p}$ to change over time and conduct online learning and inference about the allocation policy. The effectiveness of the allocation is determined by aggregating the ex-ante expected utilities using a *social welfare function* (SWF), denoted by $M(\boldsymbol{\mu} \odot \mathbf{p})$.

We consider ex-ante utilities for two reasons. First, ex-ante utilities represent expected utilities prior to allocation. In contrast, post-allocation realized utilities assign zero utility to all individuals who do not receive a resource at a given time step, which can render several commonly studied social welfare functions degenerate or uninformative under partial allocation. As a result, ex-post welfare is not a meaningful object for optimization or comparison in our setting. Second, ex-ante utilities admit a natural long-run interpretation: for a fixed allocation policy $\mathbf{p}$, the vector $\boldsymbol{\mu} \odot \mathbf{p}$ corresponds to the time-averaged utilities obtained when resources are repeatedly allocated according to $\mathbf{p}$. This interpretation is particularly important in inference-oriented settings, such as policy evaluation, sequential testing, and optimal stopping, where a single policy is deployed repeatedly and performance is assessed via long-running averages.

We consider families of SWFs satisfying three natural assumptions:

(A1) *Monotonicity*: Let $\mathbf{v}_1, \mathbf{v}_2 \in \mathbb{R}_+^n$ be two utility vectors. If $v_{1,i} \geq v_{2,i}$ for all $i \in [n]$, then $M(\mathbf{v}_1) \geq M(\mathbf{v}_2)$.

(A2) *Concavity*: $M(\mathbf{v})$ is concave in $\mathbf{v}$.

(A3) *Lipschitz continuity*: $M(\mathbf{v})$ is Lipschitz continuous in $\mathbf{v}$ w r.t. the $\ell_\infty$ norm.

Each assumption is used in our framework in a modular manner. Monotonicity is used to construct a confidence

sequence for $M(\boldsymbol{\mu} \odot \mathbf{p})$ from observed utilities (Section 4), which is our core statistical insight. Concavity enables tractable policy optimization and efficient computation of the optimal solution (Section 5.1). Finally, Lipschitz continuity supports our theoretical analysis and regret guarantees (Section 5.2) concerning the optimality of our algorithm for online SWF maximization.

## 3.1. SWF Families

We consider three popular families of SWFs.

1. *Weighted power mean (WPM)* has parameters $\mathbf{w} \in \Delta_{n-1}$ and $q \in (-\infty, 1] \cup \{-\infty\}$. It is defined as

$$M_{\text{WPM}}(\boldsymbol{\mu} \odot \mathbf{p}; \mathbf{w}, q) = \begin{cases} \min_{i \in [n]} \mu_i p_i & \text{if } q = -\infty \\ \prod_i (\mu_i p_i)^{w_i} & \text{if } q = 0 \\ \left(\sum_i w_i (\mu_i p_i)^q\right)^{1/q} & \text{otherwise} \end{cases}.$$

Intuitively, $\mathbf{w}$ encodes the relative weights given to the individuals. $q$ encodes the notion of fairness used: $q = 0$ corresponds to Nash social welfare, whereas $q = 1$ corresponds to utilitarian social welfare ($M_{\text{WPM}} = \sum_i w_i \mu_i p_i$). WPM satisfies *relative* inequality aversion (Cousins, 2023): for any $a > 0$, then $M_{\text{WPM}}(a\mathbf{u}) = a M_{\text{WPM}}(\mathbf{u})$.

2. *(Weighted) Kolm social welfare* has parameters $\mathbf{w} \in \Delta_{n-1}$ and $q \in (-\infty, 0] \cup \{-\infty\}$. It is defined as

$$M_{\text{Kolm}}(\boldsymbol{\mu} \odot \mathbf{p}; \mathbf{w}, q) = \begin{cases} \min_{i \in [n]} \mu_i p_i & \text{if } q = -\infty \\ \sum_i w_i \mu_i p_i & \text{if } q = 0 \\ \frac{1}{q} \cdot \log\left(\sum_i w_i \exp(q \mu_i p_i)\right) & \text{otherwise} \end{cases}.$$

$\mathbf{w}$ and $q$ encode individuals' relative weights and fairness notion (same as WPM). Kolm SWF satisfies the axiom of absolute inequality aversion (Kolm, 1976): for any $a > 0$ $M_{\text{Kolm}}(\mathbf{u} + a\mathbf{1}_n) = M_{\text{Kolm}}(\mathbf{u}) + a$.

3. *Gini social welfare* has parameters $\mathbf{w} \in [0,1]^n$ such that $w_1 \geq w_2 \geq \ldots \geq w_n \geq 0$, and

$$M_{\text{Gini}}(\boldsymbol{\mu} \odot \mathbf{p}; \mathbf{w}) = \sum_i w_i (\boldsymbol{\mu} \odot \mathbf{p})_{(i)},$$

where $(\boldsymbol{\mu} \odot \mathbf{p})_{(i)}$ denotes the $i$-th smallest element in the vector $\boldsymbol{\mu} \odot \mathbf{p}$. We note that with appropriate choices of $\mathbf{w}$, $M_{\text{Gini}}$ is identical to the utilitarian welfare ($\mathbf{w} = \mathbf{1}_n$) and egalitarian welfare ($w_1 = 1$, $w_i = 0$ for $i \geq 2$) with appropriate $\mathbf{w}$. The Gini SWF satisfies rank-based inequality sensitivity (Weymark, 1981), where the order of the utilities matter but not their identities.

All three SWF families span the space of social welfare formulations from egalitarian welfare to utilitarian welfare in different manners. Crucially, all three SWFs satisfy our assumptions (A1)-(A3).

**Proposition 3.1.** $M_{WPM}(\mathbf{v})$, $M_{Kolm}(\mathbf{v})$, *and* $M_{Gini}(\mathbf{v})$ *are all monotonic and concave in* $\mathbf{v}$*. Moreover,*

1. Let $u_{t,i} \in [u_{\min}, u_{\max}]$, where $u_{\max} \geq u_{\min} > 0$. Then, $M_{WPM}$ is Lipschitz continuous w.r.t. the $\ell_\infty$ norm with constant upper bounded by $u_{\max}/u_{\min}$.

2. $M_{Kolm}$ is Lipschitz continuous w.r.t. the $\ell_\infty$ norm with constant 1.

3. $M_{Gini}$ is Lipschitz continuous w.r.t the $\ell_\infty$ norm with constant $\sum_i w_i$.

We prove this result in Appendix B.1.

*Remark* 3.2. While our upper bound of $u_{\max}/u_{\min}$ for the Lipschitz constant for the WPM family holds for all $q \in (-\infty, 1] \cup \{-\infty\}$, (Cousins, 2021) provide tighter Lipschitz continuity bounds for $q < 0$ and Hölder continuity bounds for $q \in [0, 1]$. These bounds can be used for tighter theoretical guarantees (for instance, by plugging into Theorem 5.2).

We define the optimal allocation $\mathbf{p}^*$ w.r.t. the SWF $M(\cdot)$ as

$$\mathbf{p}^* = \arg \max_{\mathbf{p} \in \mathcal{P}_k} M(\boldsymbol{\mu} \odot \mathbf{p}),$$

where $\mathcal{P}_k = \{\mathbf{p} \in [0, 1]^n \mid \sum_i p_i = k\}$.

# 4. Confidence Sequence Framework Using Monotonicity

We begin by constructing a confidence sequence (CS) for the optimal welfare value $M(\boldsymbol{\mu} \odot \mathbf{p}^*)$, which will serve as the statistical backbone for learning, testing, and stopping procedures. Given observed data $\{x_t\}_{t \geq 1}$ and some $\delta \in (0, 1)$, a valid $(1 - \delta)$ CS for a target quantity $\theta$ consists of a sequence of intervals $\left\{[x_t^\downarrow, x_t^\uparrow]\right\}_{t \geq 1}$ such that $x_t^\downarrow$ and $x_t^\uparrow$ depend on the past data $\{x_s\}_{s < t}$ and $\delta$.

This sequence of intervals satisfies the property

$$\mathbb{P}(\exists t \in \mathbb{N} : \theta \notin [x_t^\downarrow, x_t^\uparrow]) \leq \delta,$$

where $\delta \in (0, 1)$. This guarantee is *time-uniform*, i.e., it holds for all time steps $t$ simultaneously. Confidence sequences are central in sequential statistical inference, enabling decision-makers to develop adaptive experiments which can be stopped flexibly. In our setting, they provide anytime-valid guarantees for welfare values induced by static or adaptively learned allocation policies, allowing hypothesis testing and stopping decisions without sacrificing statistical validity.

For instance, a decision-maker testing the hypotheses $H_0 : \theta = x_0$ versus $H_1 : \theta \neq x_0$ can reject the null as soon as $x_0 \notin [x_t^\downarrow, x_t^\uparrow]$. Time-uniform guarantees ensure that such adaptive stopping results in a valid test.

In our setting, we observe the utilities $u_{t,i}$ for the individuals to whom the resource is allocated and our target quantity is $M(\boldsymbol{\mu} \odot \mathbf{p}^*)$. CS construction is well-established for the true mean utility $\mu_i$ given the observations $\{u_{t,i}\}_{t \geq 1}$ (Howard et al., 2020; 2021). However, it is challenging to construct a CS for $M(\boldsymbol{\mu} \odot \mathbf{p}^*)$ since this quantity depends on $\boldsymbol{\mu}$ itself *and* $\mathbf{p}^*$, which is optimized given $\boldsymbol{\mu}$. Thus, we require an estimate of $\boldsymbol{\mu}$ which can be optimized over to obtain an allocation $\mathbf{p}_t$, resulting in CSs which are both valid and informative.

Our key observation is that monotonicity (A1) alone is sufficient to lift coordinate-wise confidence sequences for the individual utilities into a valid confidence sequence for the optimal social welfare. This lifting principle underlies all subsequent algorithmic and statistical guarantees.

**Theorem 4.1.** *(CS lifting) Let $\left\{\boldsymbol{\mu}_t^\downarrow\right\}_{t \geq 1}$ and $\left\{\boldsymbol{\mu}_t^\uparrow\right\}_{t \geq 1}$ be two sequences such that $[\mu_{t,i}^\downarrow, \mu_{t,i}^\uparrow]$ is a valid $(1 - \delta/n)$ confidence sequence for $\mu_i$. Moreover, let*

$$\mathbf{p}_t^\downarrow = \arg \max_{\mathbf{p} \in \mathcal{P}_k} M(\boldsymbol{\mu}_t^\downarrow \odot \mathbf{p}), \quad and$$

$$\mathbf{p}_t^\uparrow = \arg \max_{\mathbf{p} \in \mathcal{P}_k} M(\boldsymbol{\mu}_t^\uparrow \odot \mathbf{p}).$$

*Then, we have, with probability $(1 - \delta)$ uniformly,*

$$M(\boldsymbol{\mu} \odot \mathbf{p}^*) \geq M(\boldsymbol{\mu} \odot \mathbf{p}_t^\downarrow) \geq M(\boldsymbol{\mu}_t^\downarrow \odot \mathbf{p}_t^\downarrow), \quad and$$

$$M(\boldsymbol{\mu} \odot \mathbf{p}^*) \leq M(\boldsymbol{\mu}_t^\uparrow \odot \mathbf{p}^*) \leq M(\boldsymbol{\mu}_t^\uparrow \odot \mathbf{p}_t^\uparrow).$$

*Thus, $\left\{[M(\boldsymbol{\mu}_t^\downarrow \odot \mathbf{p}_t^\downarrow), M(\boldsymbol{\mu}_t^\uparrow \odot \mathbf{p}_t^\uparrow)]\right\}_{t \geq 1}$ is a valid $(1 - \delta)$ CS for $M(\boldsymbol{\mu} \odot \mathbf{p}^*)$.*

We prove this result in Appendix B.2 using only the componentwise monotonicity of $M$. This is a general result, allowing us to lift *any* valid CS for the individual mean utilities $\mu_i$ to a CS for the optimal SWF. Furthermore, when $\mathbf{p}$ is a fixed known allocation, the same construction gives $\left\{[M(\boldsymbol{\mu}_t^\downarrow \odot \mathbf{p}), M(\boldsymbol{\mu}_t^\uparrow \odot \mathbf{p})]\right\}_{t \geq 1}$ as a valid $(1 - \delta)$ CS for $M(\boldsymbol{\mu} \odot \mathbf{p})$.

Optimizing the upper bound on the lifted CS immediately yields UCB-style policies (Section 5), while the two-sided bounds support inference tasks such as sequential hypothesis testing (Section 7). We describe optimal stopping and policy evaluation as further applications in Appendix A.1.

# 5. Online SWF Maximization

We consider the task of finding the optimal allocation $\mathbf{p}^*$ in an online manner. Let $\mathbf{p}_t$ denote the allocation at time $t$ with ex-ante social welfare $M(\boldsymbol{\mu} \odot \mathbf{p}_t)$. We define regret as

$$R(T) = \sum_{t=1}^{T} \left(M(\boldsymbol{\mu} \odot \mathbf{p}^*) - M(\boldsymbol{\mu} \odot \mathbf{p}_t)\right),$$

which measures the welfare loss relative to the best fixed randomized allocation if the true mean utilities $\boldsymbol{\mu}$ were known. The stochastic multi-play multi-armed bandit setting with $k$ pulls round is retrieved in this task by considering utilitarian social welfare (achieved by all 3 of our chosen SWFs). Thus, our setting can also be interpreted as a generalization of multi-play bandits to resource allocation, with an aggregated notion of reward defined by the SWF.

## 5.1. Algorithm Using Concavity

We encounter the classic exploration-exploitation tradeoff in this task: setting a higher probability of allocation $p_{t,i}$ to individual $i$ at time $t$ gives us a better estimate of $\mu_i$. However, this is at the expense of other individuals to whom the resource could have been allocated. This is further complicated by the fact that the allocation probability $p_{t,i}$ depends on both the estimated $\mu_i$ of individual $i$ *and* the estimated mean utilities of the other individuals. Nevertheless, we develop SWF-UCB, an algorithm inspired by UCB and adapted to our setting, showing that it performs optimally.

The algorithm proceeds by constructing the policy vector $\mathbf{p}_t$ based on upper-confidence CSs of the individual mean utilities $\boldsymbol{\mu}_t^\uparrow$. A realized allocation $S_t \subseteq [n]$ is sampled using $\mathbf{p}_t$, and finally the estimates for the allocated individuals is updated using their observed utilities. Our general, SWF-agnostic algorithm is given in Algorithm 1.

---

**Algorithm 1** Generalized SWF-UCB

**Require:** Time horizon $T$, number of individuals $n$, resources per round $k$
**Require:** Initial upper confidence vector $\boldsymbol{\mu}_1^\uparrow$
1: **for** $t = 1, 2, \ldots, T$ **do**
2:    **if** $t \leq \lceil n/k \rceil$ **then**
3:      $S_t = [tk, ((t+1)k, n) \mod n + 1]$
4:    **else**
5:      *Policy optimization:*
6:        $\mathbf{p}_t \leftarrow \arg\max_{\mathbf{p} \in \mathcal{P}_k} M(\boldsymbol{\mu}_t^\uparrow \odot \mathbf{p})$
7:      *Allocation sampling:*
8:        Sample $S_t \subseteq [n]$ such that $|S_t| = k$ and $\mathbb{P}(i \in S_t) = p_{t,i}$ for all $i$
9:    **end if**
10:    *Feedback:*
11:      Observe $u_{t,i}$ for $i \in S_t$
12:    *Confidence sequence update:*
13:    **for** $i = 1, 2, \ldots, n$ **do**
14:      **if** $i \in S_t$ **then**
15:        $\mu_{t+1,i}^\uparrow \leftarrow \text{UPDATE}(\mu_{t,i}^\uparrow, u_{t,i})$
16:      **else**
17:        $\mu_{t+1,i}^\uparrow \leftarrow \mu_{t,i}^\uparrow$
18:      **end if**
19:    **end for**
20: **end for**

---

We describe the technical details of the steps below.

**Policy optimization (Line 6).** At every $t$, we solve a constrained optimization problem $\mathbf{p}_t^\uparrow = \arg\max_{\mathbf{p}} M(\boldsymbol{\mu}_t^\uparrow \odot \mathbf{p})$. Our assumption of concavity of $M(\mathbf{v})$ in $\mathbf{v}$ (A2) ensures that this objective is tractable. However, since this problem is solved at each time step, we need efficient oracles for practical feasibility. We provide exact oracles for the three SWF families we consider:

**Theorem 5.1.** *For each of the following SWF families – WPM, Kolm, and Gini – the optimization problem*

$$\mathbf{p}_t^\uparrow = \arg\max_{\mathbf{p} \in \mathcal{P}} M(\boldsymbol{\mu}_t^\uparrow \odot \mathbf{p})$$

*admits an exact oracle for the optimal solution.*

- *For WPM and Kolm, the optimal solution has a water-filling form parametrized by a scalar $\lambda$ which can be found in $\mathcal{O}(n \log n)$ time.*

- *For Gini, the optimal solution can be computed by a greedy block-based algorithm running in $\mathcal{O}(kn)$ time.*

Since WPM and Kolm SWFs are both differentiable, we employ KKT conditions to find optimal solutions. These solutions have a closed form solution parameterized by $\lambda$ which is multiplicative for WPM and additive for Kolm. We describe water-filling-based algorithms in Appendix C.1 and Appendix C.2 for WPM and Kolm SWFs, respectively.

The Gini SWF is not differentiable everywhere, making standard KKT analysis difficult. However, we prove that the permutation of individuals $\sigma$ such that $w_i$ is paired with individual $\sigma(i)$ for an optimal solution can be easily found (Proposition C.1). We developed a block-based water-filling algorithm (Appendix C.3) for this family, solving a parametric linear program similar to PAVA for isotonic regression (Best & Chakravarti, 1990).

To provide some intuition about the nature of the solutions, we consider egalitarian and utilitarian SWFs, two settings common across all three SWF families. In the utilitarian setting, the optimal solution is to allocate resources to the $k$ individuals with the highest $w_i \mu_i^\uparrow$. In the egalitarian setting, the optimal solution is $p_{t,i} \propto \left[ \lambda / \mu_{t,i}^\uparrow \right]_{[0,1]}$, where $[x]_{[0,1]}$ indicates $x$ clipped to be between 0 and 1, and $\lambda$ is chosen such that $\sum_i p_i = k$. This solution ensures that all individuals have identical values of $\mu_i^\uparrow p_i$ until the individual with the smallest $\mu_i^\uparrow$ receives a resource with $p_i = 1$.

**Allocation sampling (Line 8).** The sampled set $S_t \subseteq [n]$ should satisfy the two constraints of cardinality ($|S_t| = k$) and marginal probability ($\mathbb{P}(i \in S_t) = p_{t,i}$ for all $i$). We use *dependent rounding* (Gandhi et al., 2006), a sampling technique commonly used in survey design, to ensure that

these constraints are satisfied. We specify the algorithm for dependent rounding in Appendix D.

**Confidence update (Line 15).** We choose the sequence $\left\{\boldsymbol{\mu}_{t,i}^{\uparrow}\right\}_{t \geq 1}$ such that it is a valid CS for $\mu_i$. Our update rule is based on a well-known CS (Howard et al., 2021), which can be expressed as

$$\text{UPDATE}(\mu_{t,i}^{\uparrow}, u_{t,i}) = \frac{N_{t,i}\hat{\mu}_{t,i} + u_{t,i}}{N_{t,i} + 1}$$
$$+ 1.7\sqrt{\frac{\log(5.2\, n/\delta) + \log\log(2N_{t,i} + 1)}{N_{t,i} + 1}}, \tag{1}$$

where $N_{t,i}$ is the number of times a resource has been allocated to individual $i$ up to time $t$ and $\hat{\mu}_{t,i}$ is the empirical mean.

## 5.2. Results Using Lipschitz Continuity

We establish that Algorithm 1 achieves sub-linear regret through the following upper bound.

**Theorem 5.2.** *Let the utility distributions $\mathcal{D}_i$ be 1-sub-Gaussian for all $i$. Let $M(\mathbf{v})$ be $L$-Lipschitz continuous w.r.t. the $\ell_\infty$ norm. Let $a = 3\log(n/\delta)/2$. Then, with probability $(1 - \delta)$, for all $T \in \mathbb{N}$,*

$$R(T) \lesssim L\sqrt{\log\log T + \log(2n/\delta)}$$
$$\cdot \left( n\log\left(\frac{2n}{\delta}\right) + \sqrt{nkT - n\log\left(\frac{2n}{\delta}\right)} \right)$$

Thus, with a choice of $\delta \asymp 1/\sqrt{nkT}$, we have a bound on $\mathbb{E}[R(T)]$ of the order $\tilde{\mathcal{O}}(L(n + \sqrt{nkT})$. We use the Lipschitz continuity of $M(\cdot)$ (A3) to upper-bound the regret in terms of $\boldsymbol{\mu}_t^{\uparrow} \odot \mathbf{p}_t$ and $\boldsymbol{\mu} \odot \mathbf{p}_t$. The argument then proceeds by establishing an anytime-valid upper bound on $N_{t,i}$, which is used to establish an upper bound on the regret.

The utilitarian welfare setting corresponds to a multi-play stochastic multi-armed bandit setting, where $k$ arms are pulled at each time step simultaneously, and the total reward is the sum of the per-arm rewards. Thus, we immediately get the following lower bound on the regret $R(T)$ from prior literature (Kveton et al., 2015; Cesa-Bianchi & Lugosi, 2012):

**Proposition 5.3.** *The online SWF maximization task has a regret lower bound of $\Omega(\sqrt{nkT})$.*

Thus, our algorithm is order-optimal in $k$ and $T$ and near-optimal in $n$, matching known bounds up to poly-log factors.

There are situations where the upper bound can be made much tighter. For example, when $k = n$, every individual gets a resource at each time step, so $\mathbf{p}_t = \mathbf{p}^* = \mathbf{1}_n$, yielding

zero regret. We hypothesize that the worst-case regret is actually attained at an intermediate $k$ rather than $k = 1$ or $k = n$. Intuitively, increasing $k$ raises the cost of misallocation at given time step. However, for $k \gtrsim n/2$, the diameter of the space of possible allocations also decreases, resulting in a smaller maximum possible deviation from the optimal allocation. This leads to a non-monotonic dependence of regret on $k$. Our experiments on varying $k$ in Section 6 empirically verify this behavior.

Another such situation is the WPM SWF setting with $q = 0$. The optimal solution is of the form $\mathbf{p}_t = [\lambda w_i]_{[0,1]}$ and $\lambda$ is chosen such that $\sum_i p_{t,i} = k$. This optimal solution is independent of the estimate $\boldsymbol{\mu}_t^{\uparrow}$, thus the regret is also near-zero in this case. This indicates that although the regret bound is sub-linear and near-optimal for the worst case, there are non-trivial settings where the bound can be made tighter. We leave further analysis of special cases and the exact dependence of the regret bound on $k$ to future work (Section 8).

# 6. Experiments

Our theory provides a general, near-optimal bound for SWF-UCB under assumptions (A1)-(A3). While Theorem 5.2 establishes worst-case regret guarantees, it does not fully characterize how regret depends on key problem parameters. In our experiments, we study the empirical dependence of regret on three problem parameters: time horizon $T$, fairness (power) parameter $q$, and allocated resources $k$.

We run simulations on all three SWF families—WPM, Kolm, and Gini—for a population of $n = 50$. Individual utilities upon receiving a resource are distributed as $U_i \sim 0.1 + 0.9X_i$, where $X_i$ is Beta distributed with parameters $(\alpha_i, \beta_i)$ chosen randomly. We repeat each experiment for 5 randomly-seeded runs, holding the individual utility distributions constant and varying the randomized sampling.

We set the weight vector $\mathbf{w}$ such that $w_i \propto 0.9^{i-1}$ and $\sum_i w_i = 1$. This places greater weight on a few individuals for WPM and Kolm; for Gini, which reorders individuals by realized utility, the effect is closer to egalitarian.

**Varying horizon $T$.** We first verify that our algorithms achieve sublinear regret with the predicted $\sqrt{T}$ scaling across SWFs and allocation regimes. For all three SWFs, we consider $T$ in the range $[10^3, 2.56 \cdot 10^5]$ on a logarithmic scale. For WPM and Kolm, we set the default power parameter as $q = -2$. Figure 1 plot the regret against number of time steps with varying $k$.

For all three SWFs, we observe that $R(T)/\sqrt{T}$ is bounded above, indicating that $R(T)$ is $\mathcal{O}(\sqrt{T})$. The regret first increases and then decreases with increasing $k$. However, the relative ordering of the regret across values of $k$ changes

as $T$ grows.

**Varying power value $q$.** We next study how the fairness parameter $q$ affects learning difficulty. We observe how regret varies with the power $q$ for WPM and Kolm. We run all experiments for $T = 10^4$ time steps. We plot the variation of the regret against $q$ with different values of $k$ in Figure 2. For WPM SWF, the regret decreases until $q = 0$ and then increases until $q = 1$. The regret is relatively flat near $q = 0$ since the allocation probabilities only depend on $\mathbf{w}$. For Kolm SWF, regret increases mildly with $q$ for small $k$ ($k = 1, 5$) and decreases mildly with $q$ as $k$ grows. For $q = -\infty$, there is an increase in regret with increasing $k$, followed by a decrease.

**Varying number of resources $k$.** Finally, we examine how the number of resources $k$ affects regret. We run all experiments for $T = 10^4$ time steps. Figure 3 shows the variation of the regret against $k$ with different values of $q$.

For $q = -\infty$ in WPM and Kolm (the egalitarian case), there is an increase in the regret until $k = 20$, followed by a sharp decrease for higher $k$. As the number of resources increases in the egalitarian case, the probability of allocating a resource to the individual with the lowest predicted utility increases. However, once the individual with the lowest actual $\mu_i$ is allocated a resource with probability 1, $M(\boldsymbol{\mu} \odot \mathbf{p}_t) = \min_i \mu_i = M(\boldsymbol{\mu} \odot \mathbf{p}_t^{\uparrow})$ and the regret does not increase (since egalitarian welfare is determined by the minimum-utility individual).

For both WPM and Kolm, the regret curves flatten as $q$ increases. For Gini, the curve resembles the egalitarian case, with an increase in regret until $k = 20$ followed by a more gradual decrease. As $k$ grows, regret generally rises before falling to zero at $k = n$. This trend reflects a tradeoff: for small $k$, each sub-optimal allocation is more costly (there are fewer resources to distribute), while for $k \gtrsim n/2$, the feasible allocation space shrinks and the problem becomes easier.

Appendix E reports analogous experiments under linear weight decay. We empirically verify the $\tilde{\mathcal{O}}(\sqrt{T})$ guarantee in this regime (Figure 4)., and observe qualitatively different trends with $k$ (Figure 6) and $q$ (Figure 5), showing that the weight vector contributes nontrivially to problem difficulty.

**Summary.** Together, our experiments demonstrate three key findings. First, the $\sqrt{T}$ scaling is empirically valid across our three normatively distinct SWF families. Second, regret depends non-monotonically on $k$, with the highest regret occurring at intermediate values of $k$. Third, online welfare learning exhibits rich, structured behavior that is not present in multi-play bandits, with interactions between the SWF family, its parameters, and $k$.

## 7. Inference application: Sequential testing

The CS framework supports a range of online inference tasks. We focus here on sequential testing and cover optimal stopping and policy evaluation in Appendix A.1.

Sequential testing (Ramdas et al., 2020) for SWF-based allocation asks whether a given allocation achieves welfare greater than or equal to a value $W_0$. We consider two variants, distinguished by whether the policy vector $\mathbf{p}$ is held fixed or learned from observations.

**Fixed policy $\mathbf{p}$.** In this case, the decision-maker wants to test whether the allocation policy $\mathbf{p}$ exceeds a certain welfare value $W_0$. Here, the randomness is only in the utility samples. The hypotheses are:

$$H_0 : M(\boldsymbol{\mu} \odot \mathbf{p}) \leq W_0, \quad \text{vs} \quad H_1 : M(\boldsymbol{\mu} \odot \mathbf{p}) > W_0.$$

Let $W_{\mathbf{p},t}^{\downarrow} = M(\boldsymbol{\mu}_t^{\downarrow} \odot \mathbf{p})$, where $\boldsymbol{\mu}_t^{\downarrow}$ is a $(1 - \delta)$ lower CS for $\boldsymbol{\mu}$. Our sequential test is:

$$\text{Reject null at } \tau = \inf \left\{ t : W_{\mathbf{p},t}^{\downarrow} > W_0 \right\}.$$

We note that the lower CS $\boldsymbol{\mu}_t^{\downarrow}$ can be readily constructed by modifying the confidence update in Equation 1.

**Dynamic policy $\mathbf{p}_t$.** In this case, the decision-maker wants to know if the *optimal* allocation policy $\mathbf{p}^*$ exceeds a certain welfare value $W^*$. Here, both policy and welfare are data-dependent. The hypotheses are as follows:

$$H_0 : M(\boldsymbol{\mu} \odot \mathbf{p}^*) \leq W_0, \quad \text{vs} \quad H_1 : M(\boldsymbol{\mu} \odot \mathbf{p}^*) > W_0.$$

Let $W_t^{\downarrow} = M(\boldsymbol{\mu}_t^{\downarrow} \odot \mathbf{p}_t^{\downarrow})$, where $\boldsymbol{\mu}_t^{\downarrow}$ is a $(1 - \delta)$ lower CS for $\boldsymbol{\mu}$, and $\mathbf{p}_t^{\downarrow}$ is obtained by solving the optimization problem in Theorem 5.1.

Our sequential test is:

$$\text{Reject null at } \tau = \inf \left\{ t : W_t^{\downarrow} > W_0 \right\}.$$

Due to the time-uniform nature of the CSs above, we have

$$\mathbb{P}_{H_0}(\exists t : W_{\mathbf{p},t}^{\downarrow} > W_0) \leq \delta, \quad \text{and}$$
$$\mathbb{P}_{H_0}(\exists t : W_t^{\downarrow} > W_0) \leq \delta,$$

which ensures the validity of the sequential tests.

Since the bound is time-uniform, this testing can be adaptive, where the experiment can be stopped once enough evidence has been gathered for the null to be rejected (Ramdas et al., 2020). This idea can also be applied to test for the allocation achieving a value below a certain threshold $W_0$ by considering upper confidence sequences $\left\{ M(\boldsymbol{\mu}_t^{\uparrow} \odot \mathbf{p}_t^{\uparrow}) \right\}_{t \geq 1}$.

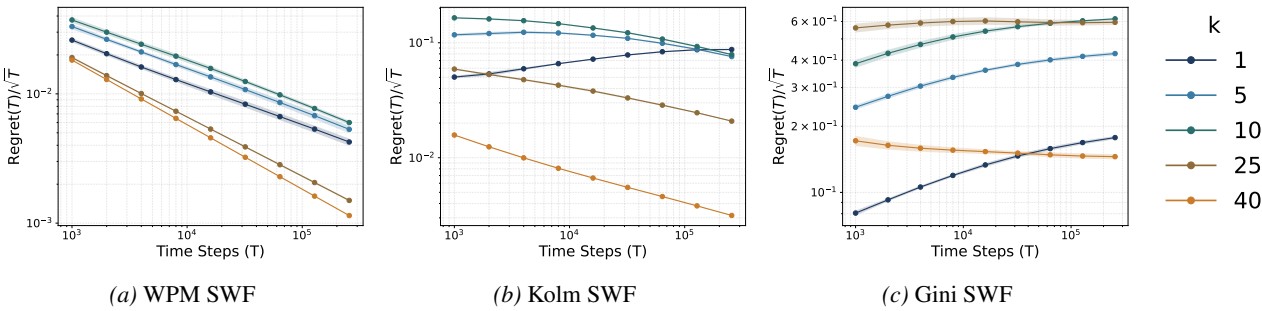

*Figure 1.* Normalized regret $R(T)/\sqrt{T}$ versus time horizon $T$ for WPM, Kolm, and Gini SWFs. The normalized regret remains bounded across two orders of magnitude in $T$, consistent with our theoretical $\tilde{\mathcal{O}}(\sqrt{T})$ guarantee.

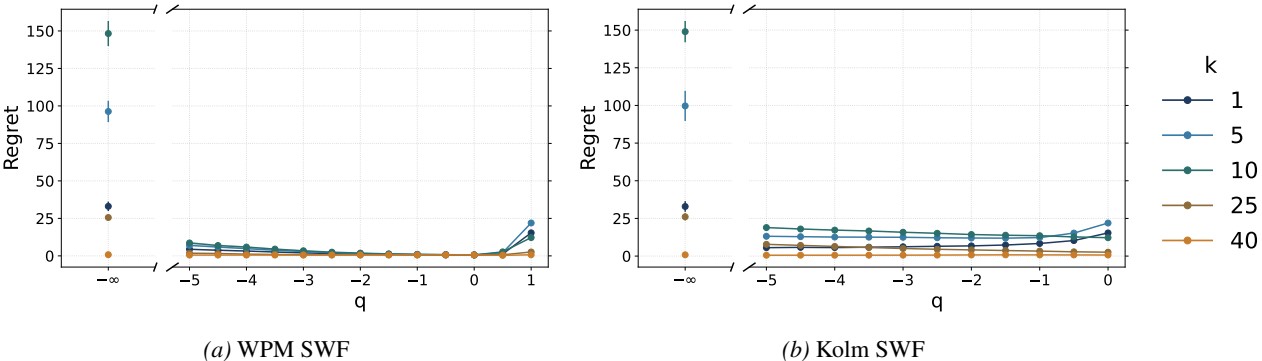

*Figure 2.* Trends with varying power value $q$ for WPM and Kolm SWFs. We consider the range between egalitarian ($q = -\infty$ for WPM and Kolm) and utilitarian ($q = 1$ for WPM and $q = 0$ for Kolm). While there is a smooth change in the observed regret, there is significant variability with changing $k$.

## 8. Discussion

**Usage of SWF families.** The three SWF families in this work encode different normative values and notions of fairness and we discuss their potential usage below.

We discuss the WPM and Kolm families jointly as they have similar theoretical formulations. The WPM family follows relative inequality aversion and is useful when *relative* differences in utility are important. The power $q \in (-\infty, 0)$ interpolates between egalitarian and Nash welfare, while $q \in (0, 1)$ interpolates between Nash and utilitarian welfare. To contrast, the Kolm family follows absolute inequality aversion, hence it can be used when *additive* differences in utility are important. The power $q \in (-\infty, 0)$ interpolates between egalitarian and utilitarian welfares. For both WPM and Kolm families,the weight vector $\mathbf{w}$ can be interpreted as encoding the relative importance of individuals in the social welfare. Thus, $\mathbf{w}$ can encode vulnerability, priority classes, or societal importance.

The Gini family is useful when positional or rank-based inequality between individuals matters: the welfare depends on who is worse-off relative to others, rather than on absolute or relative utility gaps. The weight vector $\mathbf{w}$ encodes

social priority across ranks, allowing policymakers to emphasize improvements among the bottom-ranked individuals independently of their absolute utilities.

**Direct extensions.** Our online SWF maximization 5 framework can be seamlessly used for inference applications, with Theorem 4.1 providing a valid welfare CS and the oracle algorithms in Section 5.1 providing an efficient way to learn the dynamic policy $\mathbf{p}_t^{\downarrow}$ or $\mathbf{p}_t^{\uparrow}$. While we consider sequential testing in Section 7, we explore two other applications—optimal stopping and policy evaluation—in Appendix A.1.

While we consider a fixed number of resources $k$ arriving at each time step $t$, the framework and analysis can be readily extended to accommodate a variable number of resources $k_t$ arrives at each $t$. Theorem 4.1 still holds and the oracle algorithms can be run with a different $k_t$ at each time step. In this case, we conjecture that regret guarantees would be of the form $\tilde{O}(n + \sqrt{n \sum_{t=1}^{T} k_t})$.

Finally, although we assume that the utility upon non-allocation of a resource is zero for simplicity of analysis, it should also be possible to extend the framework to situations where the non-allocation utility $u_{t,i}^{(0)}$ is distributed

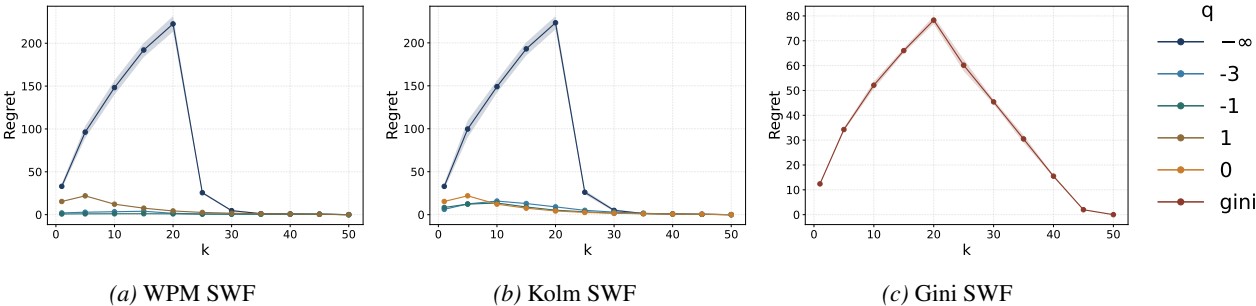

*(a)* WPM SWF  *(b)* Kolm SWF  *(c)* Gini SWF

*Figure 3.* Trends with varying number of allocated resources $k$ for all three SWFs. We observe that there is a sharp decrease after $k = 20$ for egalitarian welfare. The changes with increasing $q$ becomes less gradual for both WPM and Kolm SWFs. Geometric weights have some similarity with egalitarian welfare, and we see a similar pattern with varying $k$ for Gini SWF, although the curve is much smoother.

stochastically, and we comment on this further in A.2.

**Future work.** In Theorem 4.1, we use $(1 - \delta/n)$ CSs for the individuals' mean utilities to form a $(1 - \delta)$ CS for the social welfare. A tighter bound can be obtained by considering $(1 - \delta_i)$ CSs for the mean utilities, where $\sum_i \delta_i = \delta$, and a refined allocation of confidence across individuals. The choice of $\delta_i$ would depend on the SWF parameters and could potentially improve statistical and computational efficiency.

Theorem 5.2 provides a regret bound applicable to any SWF satisfying (A1)-(A3), showing that SWF-UCB is near-optimal in a minimax sense. However, for specific SWFs, one may obtain tighter bounds with explicit dependence on fairness parameters such as $q$ or $\mathbf{w}$, potentially improving constants and adaptivity (Remark 3.2).

Empirically, we see a non-monotonic phase transition in regret as a function of $k$: regret initially increases and then decreases as more resources are allocated. This transition depends on the SWF parameters, suggesting a rich interaction between fairness, uncertainty, and resource availability.

While we assume iid utilities, Theorem 4.1 is agnostic to the source of uncertainty and only requires valid time-uniform confidence sequences for individual utilities. This suggests that, when such CSs are available for stateful reward processes, extensions to restless (Whittle, 1988; Wang et al., 2020) or Markov (Neu et al., 2010; Ortner et al., 2012) bandits may be possible without altering the welfare-level inference machinery.

## Impact Statement

This paper develops a theoretical framework that leverages confidence sequences and social welfare functions for online resource allocation and inference. Given the abstract nature of our work, we do not anticipate that it poses a significant direct societal risk. However, the modeling and development of automated resource allocation systems raises important societal considerations. Our formulation assumes individual

utilities can be observed or estimated, yet in practice utilities are latent constructs that may be difficult to elicit, unstable, or contested. By definition, SWF parameters $(w, q)$ encode normative judgments about equity-efficiency tradeoffs that warrant stakeholder engagement through established approaches such as value-sensitive design. Additionally, fairness across protected groups may require constraints beyond SWF maximization alone. While our work is motivated by real-world allocation problems (Section 1), detailed guidance for operationalizing this framework in specific applications is beyond the current scope.

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

# A. Further Discussion

## A.1. Inference applications

Beyond sequential testing (Section 7), two further online inference applications arise naturally from the CS lifting theorem (Theorem 4.1): optimal stopping and policy evaluation. We provide their interpretations in the SWF-based allocation setting. These applications illustrate how welfare-level CSs enable valid inference under adaptive data collection.

We note that because of the generality of Theorem 4.1, other sequential inference tasks (Howard et al., 2020) can also be addressed using this framework. Moreover, in settings where $\mathbf{p}_t$ is learned dynamically, our oracle algorithms (Section 5.1) can be used to efficiently learn them.

We re-state Theorem 4.1 with a sequential inference perspective.

**Corollary A.1.** *Let* $[\boldsymbol{\mu}_t^{\downarrow}, \boldsymbol{\mu}_t^{\uparrow}]$ *be a* $(1-\delta)$ *CS for* $\boldsymbol{\mu}$.

1. *(Fixed allocation policy* $\mathbf{p}$*): Let* $\mathbf{p}$ *be a known fixed allocation policy. Let* $W_{\mathbf{p},t}^{\downarrow} = M(\boldsymbol{\mu}_t^{\downarrow} \odot \mathbf{p})$ *and* $W_{\mathbf{p},t}^{\uparrow} = M(\boldsymbol{\mu}_t^{\uparrow} \odot \mathbf{p})$. *Then* $\left\{[W_{\mathbf{p},t}^{\downarrow}, W_{\mathbf{p},t}^{\uparrow}]\right\}_{t \geq 1}$ *is a* $(1-\delta)$ *CS for* $W_{\mathbf{p}} = M(\boldsymbol{\mu} \odot \mathbf{p})$.

2. *(Dynamic allocation policy* $\mathbf{p}_t$*): Let*

$$\mathbf{p}_t^{\downarrow} = \arg \max_{\mathbf{p} \in \mathcal{P}_k} M(\boldsymbol{\mu}_t^{\downarrow} \odot \mathbf{p}) \quad and \quad \mathbf{p}_t^{\uparrow} = \arg \max_{\mathbf{p} \in \mathcal{P}_k} M(\boldsymbol{\mu}_t^{\uparrow} \odot \mathbf{p}).$$

*Let* $W_t^{\downarrow} = M(\boldsymbol{\mu}_t^{\downarrow} \odot \mathbf{p}_t^{\downarrow})$ *and* $W_t^{\uparrow} = M(\boldsymbol{\mu}_t^{\uparrow} \odot \mathbf{p}_t^{\uparrow})$. *Then* $\left\{[W_t^{\downarrow}, W_t^{\uparrow}]\right\}_{t \geq 1}$ *is a* $(1-\delta)$ *CS for* $W^* = M(\boldsymbol{\mu} \odot \mathbf{p}^*)$.

The oracle algorithms developed in Section 5.1 can be used to adaptively learn $\mathbf{p}_t^{\downarrow}$ or $\mathbf{p}_t^{\uparrow}$ efficiently. We now discuss how these CSs can be used for inference.

### A.1.1. OPTIMAL STOPPING

Consider a decision-making setting where a policy is to be learned dynamically in a testing phase. The testing phase is stopped when the social welfare achieved by the current allocation exceeds a certain value $W_0$, after which the policy is deployed. Unlike sequential testing, the goal here is not hypothesis rejection, but deciding when to transition from exploration to deployment.

Let $\mathcal{E} = \left\{\forall t \geq 1 : \boldsymbol{\mu}_t^{\downarrow} \preceq \boldsymbol{\mu}\right\}$, which holds with probability at least $(1-\delta)$. On $\mathcal{E}$, since $M$ is a monotone function, for all $t$,

$$M(\boldsymbol{\mu} \odot \mathbf{p}_t^{\downarrow}) \geq M(\boldsymbol{\mu}_t^{\downarrow} \odot \mathbf{p}_t^{\downarrow}) = W_t^{\downarrow}$$

Our stopping guarantee in this case is thus: at $\tau = \inf \left\{t : W_t^{\downarrow} > W_0\right\}$,

$$\mathbb{P}(M(\boldsymbol{\mu} \odot \mathbf{p}_\tau^{\downarrow}) > W_0) \geq 1 - \delta.$$

After stopping the experiment, we deploy $\mathbf{p}_\tau^{\downarrow}$.

### A.1.2. POLICY EVALUATION

Our method can also be used to compare the social welfare for two policies $\mathbf{p}_{(1)}$ and $\mathbf{p}_{(2)}$. This allows us to compare policies and choose the best one among them for deployment in an adaptive fashion.

Interestingly, either of these policies (or a combination of them) can be used to gather the data, provided the induced sampling process ensures coverage of all individuals (e.g. $p_{t,i} \geq \gamma > 0$ for all $i$), since we only require CSs on the individual utilities $\mu_{t,i}$. Based on the first part of Corollary A.1, we construct $(1 - \delta/2)$ welfare CSs for $\mathbf{p}_{(1)}$ and $\mathbf{p}_{(2)}$, denoting them by $\left\{[W_{(1),t}^{\downarrow}, W_{(1),t}^{\uparrow}]\right\}_{t \geq 1}$ and $\left\{[W_{(2),t}^{\downarrow}, W_{(2),t}^{\uparrow}]\right\}_{t \geq 1}$. Policy $i$ can be chosen for deployment over policy $j$ with confidence $(1 - \delta)$ when $W_{(i),t}^{\downarrow} > W_{(j),t}^{\uparrow}$.

### A.2. Extension: Stochastic Utilities for non-allocation

In Sections 3–6, we assume that the utility when an individual does not receive a resource is zero. We now discuss how the CS-based framework can be extended to stochastically observed utilities for non-allocation.

Let $u_{t,i}^{(1)} \overset{iid}{\sim} \mathcal{D}_i^{(1)}$ and $u_{t,i}^{(0)} \overset{iid}{\sim} \mathcal{D}_i^{(0)}$ be the stochastic utilities at time $t$ when an individual $i$ is allocated and not allocated a resource, respectively. Let $\mu_i^{(1)}$ and $\mu_i^{(0)}$ respectively be the mean utilities for allocation and non-allocation to individual $i$. The ex-ante utility for allocation policy $\mathbf{p}$ is thus given by

$$\overline{\boldsymbol{\mu}}(\mathbf{p}) = \boldsymbol{\mu}^{(1)} \odot \mathbf{p} + \boldsymbol{\mu}^{(0)} \odot (\mathbf{1}_n - \mathbf{p}),$$

and the social welfare is thus $M\left(\overline{\boldsymbol{\mu}}(\mathbf{p})\right)$. Thus, allocating a resource to individual $i$ yields an effective utility gain of $\mu_i^{(1)} - \mu_i^{(0)}$ relative to non-allocation.

**Extending confidence sequences.** Using the observed values, we can construct $(1 - \delta/2n)$ two-sided CSs for $\mu_i^{(1)}$ and $\mu_i^{(0)}$, denoted by $\left\{[\mu_{t,i}^{(1)\uparrow}, \mu_{t,i}^{(1)\downarrow}]\right\}_{t\geq 1}$ and $\left\{[\mu_{t,i}^{(0)\downarrow}, \mu_{t,i}^{(0)\uparrow}]\right\}_{t\geq 1}$. By the union bound, we get the following $(1 - \delta)$ CS for $\overline{\boldsymbol{\mu}}(\mathbf{p})$:

$$\left\{\left[\left(\boldsymbol{\mu}_t^{(1)\downarrow} - \boldsymbol{\mu}_t^{(0)\uparrow}\right)\mathbf{p} + \boldsymbol{\mu}_t^{(0)\downarrow}, \left(\boldsymbol{\mu}_t^{(1)\uparrow} - \boldsymbol{\mu}_t^{(0)\downarrow}\right)\mathbf{p} + \boldsymbol{\mu}_t^{(0)\uparrow}\right]\right\}_{t\geq 1}.$$

If we let

$$\mathbf{p}_t^{\downarrow} = \arg\max_{\mathbf{p}\in\mathcal{P}_{\leq k}} \left(\boldsymbol{\mu}_t^{(1)\downarrow} - \boldsymbol{\mu}_t^{(0)\uparrow}\right)\mathbf{p} + \boldsymbol{\mu}_t^{(0)\downarrow}, \quad \text{and}$$

$$\mathbf{p}_t^{\uparrow} = \arg\max_{\mathbf{p}\in\mathcal{P}_{\leq k}} \left(\boldsymbol{\mu}_t^{(1)\uparrow} - \boldsymbol{\mu}_t^{(0)\downarrow}\right)\mathbf{p} + \boldsymbol{\mu}_t^{(0)\uparrow},$$

we then have the following $(1 - \delta)$ CS:

$$\left\{\left[\left(\boldsymbol{\mu}_t^{(1)\downarrow} - \boldsymbol{\mu}_t^{(0)\uparrow}\right)\mathbf{p}_t^{\downarrow} + \boldsymbol{\mu}_t^{(0)\downarrow}, \left(\boldsymbol{\mu}_t^{(1)\uparrow} - \boldsymbol{\mu}_t^{(0)\downarrow}\right)\mathbf{p}_t^{\uparrow} + \boldsymbol{\mu}_t^{(0)\uparrow}\right]\right\}_{t\geq 1}.$$

Owing to the monotonicity of $M(\cdot)$ and proceeding through a similar sequence of inequalities as in Theorem 4.1, we get the following $(1 - \delta)$ CS for $M(\overline{\boldsymbol{\mu}}(\mathbf{p}))$

$$\left\{\left[M\left(\left(\boldsymbol{\mu}_t^{(1)\downarrow} - \boldsymbol{\mu}_t^{(0)\uparrow}\right)\mathbf{p}_t^{\downarrow} + \boldsymbol{\mu}_t^{(0)\downarrow}\right), M\left(\left(\boldsymbol{\mu}_t^{(1)\uparrow} - \boldsymbol{\mu}_t^{(0)\downarrow}\right)\mathbf{p}_t^{\uparrow} + \boldsymbol{\mu}_t^{(0)\uparrow}\right)\right]\right\}_{t\geq 1}.$$

**Extending oracle algorithms.** Our oracle algorithm has to solve a problem of the form

$$\mathbf{p}_t^{\uparrow} = \arg\max_{\mathbf{p}\in\mathcal{P}_{\leq k}} M\left(\left(\boldsymbol{\mu}_t^{(1)\uparrow} - \boldsymbol{\mu}_t^{(0)\downarrow}\right)\mathbf{p} + \boldsymbol{\mu}_t^{(0)\uparrow}\right).$$

For some individual $i$, if $\mu_{t,i}^{(0)\downarrow} > \mu_{t,i}^{(1)\uparrow}$, we can infer with high confidence that not allocating a resource gives them higher utility. In this case, allocating a resource to individual $i$ can only decrease the welfare objective with high probability, and monotonicity implies that setting $p_i = 0$ is optimal. Thus, such individuals can be excluded from the allocation, and the optimization problem can be solved only for the remaining individuals. This in turn raises the possibility of less than $k$ resources being allocated to the population, which results in our feasible set for $\mathbf{p}$ becoming

$$\mathcal{P}_{\leq k} = \left\{\mathbf{p} \in [0, 1]^n : \sum_i p_i = s, s \in \mathbb{N}, s \leq k\right\}$$

## B. Proofs

### B.1. Proof of Proposition 3.1

*Proof. Monotonicity and concavity*: Monotonicity is guaranteed by the axioms of social welfare for $M_{\text{WPM}}$, $M_{\text{Kolm}}$, and $M_{\text{Gini}}$. We now establish concavity of the three SWF families.

- *WPM*: We begin by noting that for $q = -\infty$, $M_{\text{WPM}}(\mathbf{v}) = \min_i v_i$ is concave in $\mathbf{v}$, since for two valid vectors $\mathbf{v}_1$ and $\mathbf{v}_2$,

$$\min_i \lambda v_{1,i} + (1 - \lambda)v_{2,i} \geq \lambda \min_i v_{1,i} + (1 - \lambda) \min_j v_{2,j}$$

For $q \in (-\infty, 1)$, we express the WPM SWF as

$$M_{\text{WPM}}(\mathbf{v}; \mathbf{w}; q) = \left( \sum_i \left( w_i^{1/q} v_i \right)^q \right)^{1/q}$$

Let $\mathbf{v}_1$ and $\mathbf{v}_2$ be two valid vectors. Applying the reverse Minkowski inequality, we get

$$M_{\text{WPM}}(\lambda \mathbf{v}_1 + (1 - \lambda)\mathbf{v}_2; \mathbf{w}, q) = \left( \sum_i \left( \lambda w_i^{1/q} v_{1,i} + (1 - \lambda) w_i^{1/q} v_{2,i} \right)^q \right)^{1/q}$$

$$\geq \left( \sum_i \left( \lambda w_i^{1/q} v_{1,i} \right)^q \right)^{1/q} + \left( \sum_i \left( (1 - \lambda) w_i^{1/q} v_{w,i} \right)^q \right)^{1/q}$$

$$= \lambda M_{\text{WPM}}(\mathbf{v}_1; \mathbf{w}, q) + (1 - \lambda) M_{\text{WPM}}(\mathbf{v}_2; \mathbf{w}, q)$$

- *Kolm*: Since $q = -\infty$ is the egalitarian case and $q = 0$ is the utilitarian case, they have already been shown to be concave via the WPM SWF. For $q \in (-\infty, 0)$, the concavity of $M_{\text{Kolm}}$ follows from the log-sum-exp function being convex, and it being pre-multiplied by $1/q$ (a negative quantity).

- *Gini*: The Gini SWF can also be expressed as

$$M_{\text{Gini}}(\mathbf{v}; \mathbf{w}) = \min_\sigma \sum_i w_{\sigma(i)} v_i,$$

where $\sigma$ is a permutation of the set $\{1, \dots, n\}$. Since we are taking the minimum of a set of affine functions, the resultant is a concave function.

*Lipschitz continuity*: We now establish the Lipschitz continuity of the SWF families. By the mean value theorem we know that for two valid vectors $\mathbf{v}_1$ and $\mathbf{v}_2$, there is a vector $\mathbf{x}$ such that

$$M(\mathbf{v}_1) - M(\mathbf{v}_2) = \nabla M(\mathbf{x}) \cdot (\mathbf{v}_1 - \mathbf{v}_2)$$

$$|M(\mathbf{v}_1) - M(\mathbf{v}_2)| = |\nabla M(\mathbf{x}) \cdot (\mathbf{v}_1 - \mathbf{v}_2)| \overset{(i)}{\leq} \|\nabla M(\mathbf{x})\|_1 \cdot \|\mathbf{v}_1 - \mathbf{v}_2\|_\infty,$$

where $(i)$ uses Hölder's inequality. Thus, to show Lipschitz continuity w.r.t. the $\ell_\infty$ norm, we provide bounds on $\|\nabla M(\mathbf{x})\|_1$

- *WPM*: Let $v_i \in [v_{\min}, v_{\max}]$ for $v_{\max} \geq v_{\min} > 0$. We then have, for $q \in (-\infty, 0) \cup (0, 1]$,

$$M_{\text{WPM}}(\mathbf{v}; \mathbf{w}, q) = \left( \sum_i w_i v_i^q \right)^{1/q}$$

$$\frac{\partial}{\partial v_i} M_{\text{WPM}}(\mathbf{v}; \mathbf{w}, q) = \frac{1}{q} \cdot \left( \sum_i w_i v_i^q \right)^{1/q - 1} \cdot q w_i v_i^{q-1}$$

$$= M_{\text{WPM}}(\mathbf{v}; \mathbf{w}, q) \cdot \frac{w_i v_i^q}{\sum_i w_i v_i^q} \cdot \frac{1}{v_i}$$

$$\implies \sum_i \left| \frac{\partial}{\partial v_i} M_{\text{WPM}}(\mathbf{v}; \mathbf{w}, q) \right| \leq \frac{M}{v_{\min}} \leq \frac{v_{\max}}{v_{\min}}.$$

For $q = 0$, we have

$$M_{\text{WPM}}(\mathbf{v}; \mathbf{w}, 0) = \prod_i v_i^{w_i}$$

$$\frac{\partial}{\partial v_i} = \prod_i v_i^{w_i} \cdot \frac{w_i}{v_i} = M(\mathbf{v}; \mathbf{w}, 0) \cdot \frac{w_i}{v_i}$$

$$\implies \sum_i \left| \frac{\partial}{\partial v_i} M_{\text{WPM}}(\mathbf{v}; \mathbf{w}, q) \right| \leq \frac{M}{v_{\min}} \leq \frac{v_{\max}}{v_{\min}}.$$

For $q = -\infty$, we observe that the sum of all subgradients is upper bounded by 1, which is clearly less than $v_{\max}/v_{\min}$. Thus, $v_{\max}/v_{\min}$ is the Lipschitz constant.

We additionally note that for $q = 1$, $M_{\text{WPM}}(\mathbf{v}; \mathbf{w}, 1) = \sum_i w_i v_i$ has $\|\nabla M_{\text{WPM}}(\mathbf{v}; \mathbf{w}, 1)\|_1 = \sum_i w_i = 1$.

- *Kolm*: The case of $q = -\infty$ (egalitarian) and $q = 0$ (utilitarian) is already considered in the WPM case. For $q \in (-\infty, 0)$, we have

$$M_{\text{Kolm}}(\mathbf{v}; \mathbf{w}, q) = \frac{1}{q} \log \left( \sum_i \exp(qv_i) \right)$$

$$\frac{\partial}{\partial v_i} M_{\text{Kolm}}(\mathbf{v}; \mathbf{w}, q) = \frac{1}{q} \cdot \frac{1}{\sum_i w_i \exp(qu_i)} \cdot w_i q \exp(qu_i) = \frac{w_i \exp(qu_i)}{\sum_i w_i \exp(qu_i)}$$

$$\implies \sum_i \left| \frac{\partial}{\partial v_i} M_{\text{Kolm}}(\mathbf{v}; \mathbf{w}, q) \right| = 1$$

- *Gini*: We consider the alternate expression for the Gini SWF:

$$M_{\text{Gini}}(\mathbf{v}; \mathbf{w}) = \min_\sigma \sum_i w_{\sigma(i)} v_i.$$

We note that this is the minimum over affine functions $f_\sigma(\mathbf{v}) = \sum_i w_{\sigma_i} v_i$, each of which has $\|\nabla f_\sigma\|_1 = \sum_i w_i = L$. Thus, the subgradient of $M_{\text{Gini}}$ is such that $\|\partial M_{\text{Gini}}(\mathbf{v}; \mathbf{w})\|_1 \leq L$.

$\square$

## B.2. Proof of Theorem 4.1

*Proof.* For all $i \in [n]$, let $\left\{ [\mu_{t,i}^\downarrow, \mu_{t,i}^\uparrow] \right\}_{t \geq 1}$ be a $(1 - \delta/n)$ confidence sequence for $i$. That is,

$$\mathbb{P} \left( \exists t \in \mathbb{N} : \mu_i \notin [\mu_{t,i}^\downarrow, \mu_{t,i}^\uparrow] \right) \leq \frac{\delta}{n}$$

By the union bound, we thus have that for the sequence $\left\{ [\boldsymbol{\mu}_t^\downarrow, \boldsymbol{\mu}_t^\uparrow] \right\}_{t \geq 1}$

$$\mathbb{P} \left( \exists t \in \mathbb{N} : \boldsymbol{\mu} \notin [\boldsymbol{\mu}_t^\downarrow, \boldsymbol{\mu}_t^\uparrow] \right) \leq \delta,$$

where $\mathbf{x} \in [\mathbf{x}_t^\downarrow, \mathbf{x}_t^\uparrow]$ means that $x_i \in [x_{t,i}^\downarrow, x_{t,i}^\uparrow]$ for all $i \in [n]$.

Thus, with probability $(1 - \delta)$ we have

$$M(\boldsymbol{\mu} \odot \mathbf{p}^*) \overset{(i)}{\geq} M(\boldsymbol{\mu} \odot \mathbf{p}_t^\downarrow) \overset{(ii)}{\geq} M(\boldsymbol{\mu}_t^\downarrow \odot \mathbf{p}_t^\downarrow), \quad \text{and}$$

$$M(\boldsymbol{\mu} \odot \mathbf{p}^*) \overset{(iii)}{\leq} M(\boldsymbol{\mu}_t^\uparrow \odot \mathbf{p}^*) \overset{(iv)}{\leq} M(\boldsymbol{\mu}_t^\uparrow \odot \mathbf{p}_t^\uparrow).$$

$(i)$ results from $\mathbf{p}^*$ being optimal for $\boldsymbol{\mu}$, and $(ii)$ results from the CS bound and the monotonicity of $M(\cdot)$. $(iii)$ comes from the CS bound and the monotonicity of $M(\cdot)$, and $(iv)$ comes from $\mathbf{p}_t^\uparrow$ being optimal for $\boldsymbol{\mu}_t^\uparrow$.

This means that

$$\mathbb{P}(\exists t \in \mathbb{N} : M(\boldsymbol{\mu} \odot \mathbf{p}^*) \notin [M(\boldsymbol{\mu}_t^\downarrow \odot \mathbf{p}_t^\downarrow), M(\boldsymbol{\mu}_t^\uparrow \odot \mathbf{p}_t^\uparrow)]) \leq \delta$$

$\square$

## B.3. Proof of Theorem 5.2

*Proof.* Let $\left\{\boldsymbol{\mu}_t^\uparrow\right\}_{t \geq 1}$ be a $(1 - \delta/2)$ confidence sequence for $\boldsymbol{\mu}$. From Theorem 4.1, we know that, with probability $(1 - \delta/2)$

$$
M(\boldsymbol{\mu} \odot \mathbf{p}^*) \overset{(iii)}{\leq} M(\boldsymbol{\mu}_t^\uparrow \odot \mathbf{p}^*) \overset{(iv)}{\leq} M(\boldsymbol{\mu}_t^\uparrow \odot \mathbf{p}_t^\uparrow).
$$

Thus, we have, with probability $(1 - \delta/2)$,

$$
\begin{aligned}
R(T) &= \sum_{t=1}^T M(\boldsymbol{\mu} \odot \mathbf{p}^*) - M(\boldsymbol{\mu} \odot \mathbf{p}_t^\uparrow) \\
&\leq \sum_{t=1}^T M(\boldsymbol{\mu}_t^\uparrow \odot \mathbf{p}^*) - M(\boldsymbol{\mu} \odot \mathbf{p}_t^\uparrow) \\
&\leq \sum_{t=1}^T M(\boldsymbol{\mu}_t^\uparrow \odot \mathbf{p}_t^\uparrow) - M(\boldsymbol{\mu} \odot \mathbf{p}_t^\uparrow)
\end{aligned}
$$

If we now assume that $M$ is $L$-Lipschitz w.r.t. $\|\cdot\|_\infty$ (the maximum value), we get

$$
\begin{aligned}
R(T) &\leq \sum_{t=1}^T M(\boldsymbol{\mu}_t^\uparrow \odot \mathbf{p}_t^\uparrow) - M(\boldsymbol{\mu} \odot \mathbf{p}_t^\uparrow) \\
&\leq \sum_{t=1}^T L \left\| \boldsymbol{\mu}_t^\uparrow \odot \mathbf{p}_t^\uparrow - \boldsymbol{\mu} \odot \mathbf{p}_t^\uparrow \right\|_\infty
\end{aligned}
$$

Thus, we have bounded the regret in terms of the confidence sequence for $\boldsymbol{\mu}$. By Hölder's inequality, we thus have

$$
\begin{aligned}
\sum_{t=1}^T L \|\boldsymbol{\mu}_t^\uparrow \odot \mathbf{p}_t^\uparrow - \boldsymbol{\mu} \odot \mathbf{p}_t^\uparrow\|_\infty &\leq \sum_{t=1}^T L \|\boldsymbol{\mu}_t^\uparrow \odot \mathbf{p}_t^\uparrow - \boldsymbol{\mu} \odot \mathbf{p}_t^\uparrow\|_1 \\
&= L \sum_{t=1}^T \sum_{i=1}^n |\mu_{t,i}^\uparrow - \mu_i| p_{t,i}^\uparrow \\
&\lesssim L \sum_{t=1}^T \sum_{i=1}^n p_{t,i} \sqrt{\frac{\log\log(N_{t,i} + 1) + \log(n/\delta)}{N_{t,i} + 1}} \\
&\lesssim L \sqrt{\log\log T + \log(n/\delta)} \underbrace{\sum_{i=1}^n \sum_{t=1}^T \frac{p_{t,i}}{\sqrt{N_{t,i} + 1}}}_{S}
\end{aligned}
$$

We now bound the term $S$. We observe that $N_{t,i}$ is the sum of Bernoulli random variables $N_{t,i} = \sum_{s=1}^t X_{s,i}$, where $X_{s,i} \sim \mathrm{Ber}(p_{s,i})$. We then have the following anytime-valid guarantee (from the Master theorem in (Howard et al., 2020)):

$$
\mathbb{P}\left(\exists t \in \mathbb{N} : \sum_{s=1}^t (p_{s,i} - X_{s,i}) \geq a + b \sum_{s=1}^t p_{s,i}\right) \leq \exp\left(-\frac{2ab}{1+b}\right)
$$

We set $b = 0.5$, and $a = (3/2) \cdot \log(2n/\delta)$, so that the upper bound above is $\delta/2n$. We then have that with probability $(1 - \delta/2n)$, for all $t \in \mathbb{N}$,

$$
\begin{aligned}
&\sum_{s=1}^t (p_{s,i} - X_{s,i}) \leq a + b \sum_{s=1}^t p_{s,i} \\
&\implies N_{t,i} = \sum_{s=1}^t X_{s,i} \geq 0.5 \sum_{s=1}^t p_{s,i} - a
\end{aligned}
$$

We can now consider two cases:

1. $\sum_{s=1}^{t} p_{s,i} \leq 2a$: Here, we can say

$$\sum_{s=1}^{t} \frac{p_{s,i}}{\sqrt{N_{s,i}+1}} \leq \sum_{s=1}^{t} p_{s,i} \leq 2a,$$

which is a constant. Thus in this case, the regret is upper-bounded by a constant.

2. $\sum_{s=1}^{t} p_{s,i} > 2a$: Let this be true for $t \geq t_0$. We then have, with probability $(1 - 2\delta)$,

$$\sum_{s=t_0}^{t} \frac{p_{s,i}}{\sqrt{N_{s,i}+1}} \leq \sum_{s=t_0}^{t} \frac{1}{\sqrt{0.5(\sum_{k=1}^{s} p_{k,i}) - a}}$$

$$\leq 2\sqrt{2} \cdot \sqrt{\left(\sum_{s=1}^{t} p_{s,i}\right) - 2a}$$

We thus have

$$\sum_{t=1}^{T} \frac{p_{s,i}}{\sqrt{N_{s,i}+1}} \lesssim a + \sqrt{\left(\sum_{t=1}^{T} p_{t,i}\right) - a}$$

$$\implies \sum_{i=1}^{n} \sum_{t=1}^{T} \frac{p_{t,i}}{\sqrt{N_{t,i}+1}} \lesssim na + \sum_{i=1}^{n} \sqrt{\left(\sum_{t=1}^{T} p_{t,i}\right) - a}$$

$$\leq na + \sqrt{nkT - na}$$

Thus, we have

$$R(T) \lesssim L\sqrt{\log\log T + \log\left(\frac{2n}{\delta}\right)} \left[n\log\left(\frac{2n}{\delta}\right) + \sqrt{nkT - n\log\left(\frac{2n}{\delta}\right)}\right]$$

We thus get a rate of the form $\mathcal{O}(\sqrt{\log\log T + \log n}(n\log n + \sqrt{nkT}))$. $\qquad\square$

## C. Details of Theorem 5.1

### C.1. WPM SWF

#### C.1.1. DERIVING PROPOSED SOLUTION

Our objective can be stated as

$$\text{maximize } M_{\text{WPM}}(\mathbf{u} \odot \mathbf{p}; \mathbf{w}, q) = \left(\sum_{i} w_i(u_i p_i)^q\right)^{1/q}$$

$$\text{s.t. } \sum_{i} p_i = k$$

$$p_i - 1 \leq 0$$

$$-p_i \leq 0$$

First, let us consider the case where the $p_i$'s are unrestricted. After differentiation w.r.t. $p_i$, at the optimal point, we would want

$$\frac{\partial}{\partial p_i} M_{\text{WPM}} = \lambda'$$

$$\lambda' = \frac{1}{q} \cdot \left(\sum_{i} w_i(u_i p_i)^q\right)^{(1/q-1)} \cdot w_i u_i^q \cdot q p_i^{q-1}$$

$$= M_{\text{WPM}}(\mathbf{u} \odot \mathbf{p}; \mathbf{w}, q) \cdot \frac{w_i u_i^q}{\sum_{i} w_i(u_i p_i)^q} p_i^{q-1}$$

$$p_i = \lambda(w_i u_i^q)^{1/(1-q)},$$

where

$$\lambda = \left( \frac{\lambda' \cdot \sum_i w_i(u_i p_i^q)}{M_{\text{WPM}}(\mathbf{u} \odot \mathbf{p}; \mathbf{w}, q)} \right)^{1/(q-1)}$$

Our proposed solution is thus

$$p_i = \left[ \lambda(w_i u_i^q)^{1/(1-q)} \right]_{[0,1]}, \tag{2}$$

which is $p_i$ restricted to be between 0 and 1, with $\lambda$ chosen such that $\sum_i p_i = k$.

### C.1.2. PROVING OPTIMALITY VIA KKT CONDITIONS

The Lagrangian for the objective is

$$\mathcal{L}(\mathbf{p}, \boldsymbol{\alpha}, \boldsymbol{\beta}, \gamma) = -M_{\text{WPM}}(\mathbf{u} \odot \mathbf{p}; \mathbf{w}, q) + \sum_i \alpha_i(p_i - 1) + \sum_i \beta_i(-p_i) + \gamma(\sum_i p_i - k)$$

Recall that KKT conditions require the following:

1. *Stationarity*: For all $i$,

$$\frac{\partial}{\partial p_i}\mathcal{L} = -M_{\text{WPM}}(\mathbf{u} \odot \mathbf{p}; \mathbf{w}, q) \cdot \frac{w_i(u_i p_i)^q}{\sum_i w_i(u_i p_i)^q} \cdot \frac{1}{p_i} + \alpha_i - \beta_i + \gamma = 0$$

2. *Primal feasibility*: $\sum_i p_i = k$, and $p_i \in [0, 1]$ for all $i$.

3. *Dual feasibility*: $\alpha_i \geq 0$, $\beta_i \geq 0$ for all $i$.

4. *Complementary slackness*: $\alpha_i(1 - p_i) = 0$ and $\beta_i p_i = 0$ for all $i$.

We choose our proposed solution $\mathbf{p}^*$ in Equation 2 such that $0 < p_i \leq 1$, and $\sum_i p_i = k$. This immediately tells us that this solution is primal feasible; moreover, due to complementary slackness, we know that $\beta_i = 0$ for all $i$.

We now consider two cases for $\alpha_i$:

(i) If $p_i < 1$, complementary slackness requires $\alpha_i = 0$. In this case, we have

$$p_i^* = \lambda(w_i u_i^q)^{1/(1-q)}$$
$$u_i p_i^* = \lambda(w_i u_i)^{1/(1-q)}$$

Stationarity requires

$$\frac{\partial}{\partial p_i}\mathcal{L} = -M_{\text{WPM}}(\mathbf{u} \odot \mathbf{p}; \mathbf{w}, q) \cdot \frac{w_i(u_i p_i)^q}{\sum_i w_i(u_i p_i)^q} \cdot \frac{1}{p_i} + \alpha_i - \beta_i + \gamma = 0$$

$$M_{\text{WPM}}(\mathbf{u} \odot \mathbf{p}; \mathbf{w}, q) \cdot \frac{w_i(u_i p_i)^q}{\sum_i w_i(u_i p_i)^q} \cdot \frac{1}{p_i} = \gamma$$

$$\frac{M_{\text{WPM}}(\mathbf{u} \odot \mathbf{p}; \mathbf{w}, q)}{\sum_i w_i(u_i p_i)^q} u_i w_i(u_i p_i)^{q-1} = \gamma$$

$$\frac{M_{\text{WPM}}(\mathbf{u} \odot \mathbf{p}; \mathbf{w}, q)}{\sum_i w_i(u_i p_i)^q} u_i w_i(\lambda(w_i u_i)^{1/(1-q)})^{q-1} = \gamma$$

$$\frac{M_{\text{WPM}}(\mathbf{u} \odot \mathbf{p}; \mathbf{w}, q)}{\sum_i w_i(u_i p_i)^q} \frac{u_i w_i}{u_i w_i} \lambda^{q-1} = \gamma$$

$$\frac{M_{\text{WPM}}(\mathbf{u} \odot \mathbf{p}; \mathbf{w}, q)}{\sum_i w_i(u_i p_i)^q} \lambda^{q-1} = \gamma$$

Since the LHS does not depend on $i$, we can set $\gamma$ to be the LHS to satisfy stationarity.

(ii) If $p_i = 1$, we have $1 = p_i < \lambda(w_i u_i^q)^{1/(1-q)}$. Thus,

$$1 \leq \lambda(w_i u_i^q)^{1/(1-q)}$$

$$1 \stackrel{(i)}{\leq} \lambda^{1-q} w_i u_i^q$$

$$\lambda^{q-1} \stackrel{(i)}{\leq} w_i u_i^q$$

$$\lambda^{q-1} \leq \frac{w_i(u_i p_i)^q}{p_i}$$

$$\frac{M_{\text{WPM}}(\mathbf{u} \odot \mathbf{p}; \mathbf{w}, q)}{\sum_i w_i(u_i p_i)^q} \lambda^{q-1} \stackrel{(i)}{\leq} M_{\text{WPM}}(\mathbf{u} \odot \mathbf{p}; \mathbf{w}, q) \cdot \frac{w_i(u_i p_i)^q}{\sum_i w_i(u_i p_i)^q} \cdot \frac{1}{p_i}$$

$$\gamma \leq M_{\text{WPM}}(\mathbf{u} \odot \mathbf{p}; \mathbf{w}, q) \cdot \frac{w_i(u_i p_i)^q}{\sum_i w_i(u_i p_i)^q} \cdot \frac{1}{p_i}$$

where we set $p_i = 1$ in $(i)$, and set $\gamma$ from the first case in $(ii)$. Stationarity requires

$$\frac{\partial}{\partial p_i} \mathcal{L} = -M_{\text{WPM}}(\mathbf{u} \odot \mathbf{p}; \mathbf{w}, q) \cdot \frac{w_i(u_i p_i)^q}{\sum_i w_i(u_i p_i)^q} \cdot \frac{1}{p_i} + \alpha_i + \gamma = 0$$

This means we have $\alpha_i \geq 0$, which establishes dual feasibility.

Thus, all necessary conditions hold for $\mathbf{p}^*$, indicating that it is a local maximum. Since we are maximizing a concave function on a convex polytope, this would also be the global maximum.

**Limiting cases.** We separately comment on three limiting values of $q$ in this setting:

1. $q = 1$ (weighted utilitarian): Here, the limiting solution results in the allocation probability $p_i = 1$ for the $k$ individuals with the highest $w_i \mu_i$, resulting in a probability of $p_i = 0$ for the other individuals.

2. $q = 0$ (weighted Nash): Here, the limiting solution is $p_i = [\lambda w_i]_{[0,1]}$, where $\lambda$ is chosen such that $\sum_i p_i = 1$. Crucially, this solution does not depend on the utility vector $\boldsymbol{\mu}$; thus, the optimal solution can be derived without any observations.

3. $q = -\infty$ (egalitarian): Here, the limiting solution is $p_i = [\lambda/\mu_i]_{[0,1]}$, where $\lambda$ is chosen such that $\sum_i p_i = 1$.

These limiting cases arise as pointwise limits of the objective. Since the feasible set is compact and the objective is concave for all finite $q$, standard continuity arguments imply that the closed-form solutions converge to optimal solutions of the limiting problems.

### C.1.3. ALGORITHM

We describe the algorithm for the WPM allocation oracle in Algorithm 2. Intuitively, the algorithm proceeds by calculating filling rates for allocation probabilities over the individuals. When the fastest-filling individual $i$ reaches $p_i = 1$, that individual is excluded from further allocation. This proceeds until $\sum_i p_i = k$. The sorting of the water-filling rates (Line 15) dominates the time complexity, resulting in an $\mathcal{O}(n \log n)$ total complexity.

## C.2. Kolm SWF

### C.2.1. DERIVING PROPOSED SOLUTION

We want to optimize the Kolm social welfare under constraints. The problem is given by, for $q < 0$,

$$
\begin{aligned}
\text{maximize} \quad & M_{\text{Kolm}}(\mathbf{u}; \mathbf{w}, q) = \frac{1}{q} \log \left( \sum_i w_i \exp(q u_i p_i) \right) \\
\text{s.t.} \quad & p_i \in [0, 1] \quad \forall i \\
& \sum_i p_i = k,
\end{aligned}
$$

---

**Algorithm 2** Weighted Power Mean (WPM) Allocation Solver

---

1: **Input:** Utilities $u \in \mathbb{R}^n$, weights $w$, power parameter $q \leq 1$, resources $k$
2: **Output:** Optimal allocation policy $p \in [0,1]^n$
3: **// Step 1: Calculate individual allocation rates**
4: **if** $q = -\infty$ (Egalitarian) **then**
5:    $r_i \leftarrow 1/u_i$ for all $i$
6: **else if** $q = 0$ (Nash) **then**
7:    $r_i \leftarrow w_i$ for all $i$
8: **else if** $q = 1$ (Utilitarian) **then**
9:    $p \leftarrow$ set 1.0 for $k$ indices with largest $(w_i \cdot u_i)$, else 0.0; **return** $p$
10: **else**
11:    $\text{log\_rates}_i \leftarrow \frac{\ln(w_i) + q \cdot \ln(u_i)}{1-q}$ for all $i$
12:    $r \leftarrow \text{Softmax}(\text{log\_rates})$ {Ensures $\sum r_i = 1$}
13: **end if**
14: **// Step 2: Water-filling across sorted containers**
15: Sort $r$ in descending order: $r_{(1)} \geq r_{(2)} \geq \cdots \geq r_{(n)}$ and track indices $\pi$
16: Initialize $p \leftarrow \mathbf{0}$, $rem \leftarrow k$
17: Precompute suffix sums $R_i = \sum_{j=i}^n r_{(j)}$
18: **for** $i = 1$ to $n$ **do**
19:    $t \leftarrow 1/r_{(i)}$ {Time required to fill current container to 1.0}
20:    **if** $t \cdot R_i > rem$ **then**
21:      $t_{final} \leftarrow rem/R_i$ {Remaining mass doesn't fill any more containers fully}
22:      $p_{\pi(j)} \leftarrow t_{final} \cdot r_{(j)}$ for $j \in \{i, \ldots, n\}$
23:      $rem \leftarrow 0$; **break**
24:    **else**
25:      $p_{\pi(i)} \leftarrow 1.0$
26:      $rem \leftarrow rem - 1$
27:    **end if**
28: **end for**
29: **return** $p$

---

where $k \in [n]$.

First, let us consider the case where the $p_i$'s are unrestricted. After differentiation w.r.t. $p_i$, at the optimal point, we would want

$$\frac{\partial}{\partial p_i} M_{\text{Kolm}} = \lambda$$

$$\lambda = \frac{1}{q} \cdot \frac{1}{\sum_i w_i \exp(q u_i p_i)} \cdot w_i q u_i \exp(q u_i p_i)$$

$$p_i = \log\left(\frac{\lambda'}{w_i u_i}\right) \cdot \frac{1}{q u_i},$$

where $\lambda' = \lambda \cdot (\sum_i w_i \exp(q u_i p_i))$.

The general solution for Kolm SWF is thus

$$p_i = \left[\log\left(\frac{\lambda'}{w_i u_i}\right) \cdot \frac{1}{q u_i}\right]_{[0,1]}$$

$$= \left[\frac{\eta + \log(w_i u_i)}{|q| u_i}\right]_{[0,1]},$$

where $\eta = -\log \lambda'$, and $p_i$ is clamped to $[0,1]$ such that $\sum_i p_i = k$. This is a harder problem than the WPM case, since here $\eta = \log \lambda'$ is *additive* rather than *multiplicative*. Thus both constraints $p_i \geq 0$ and $p_i \leq 1$ might be active. We think of water-filling in terms of $\eta$.

C.2.2. PROVING OPTIMALITY THROUGH KKT CONDITIONS

We note that

$$M_{\text{Kolm}}(\mathbf{u} \odot \mathbf{p}; \mathbf{w}, q) = \frac{1}{q} \log \left( \sum_i w_i \exp(q u_i p_i) \right),$$

and

$$\frac{\partial}{\partial p_i} M_{\text{Kolm}}(\mathbf{u} \odot \mathbf{p}; \mathbf{w}, q) = \frac{1}{q} \cdot \frac{w_i \exp(q u_i p_i)}{\sum_i w_i \exp(q u_i p_i)} \cdot q u_i$$

$$= \frac{w_i \exp(q u_i p_i)}{\sum_i w_i \exp(q u_i p_i)} \cdot u_i$$

Since $q \leq 0$, we note that $\partial M_{\text{Kolm}} / \partial p_i$ is a non-increasing function of $p_i$.

Our optimization objective is:

$$\text{maximize} \quad \frac{1}{q} \log \left( \sum_i w_i \exp(q u_i p_i) \right)$$

$$\text{s.t.} \quad -p_i \leq 0 \quad \forall i$$

$$p_i - 1 \leq 0 \quad \forall i$$

$$\sum_i p_i = k$$

The Lagrangian for this expression with the optimal solution $\mathbf{p}^*$ is

$$\mathcal{L} = -M_K(\mathbf{u} \odot \mathbf{p}^*; \mathbf{w}, q) - \sum_i \alpha_i p_i^* + \sum_i \beta_i (p_i^* - 1) + \gamma (\sum_i p_i^* - k)$$

The partial derivative w.r.t. $p_i$ is

$$\left. \frac{\partial}{\partial p_i} \mathcal{L} \right|_{\mathbf{p}=\mathbf{p}^*} = -\frac{w_i \exp(q u_i p_i^*)}{\sum_i w_i \exp(q u_i p_i^*)} \cdot u_i - \alpha_i + \beta_i + \gamma$$

The KKT conditions require:

1. *Stationarity*: $0 \in \partial \mathcal{L} / \partial p_i$, which implies that

$$0 = -\frac{w_i \exp(q u_i p_i^*)}{\sum_i w_i \exp(q u_i p_i^*)} \cdot u_i - \alpha_i + \beta_i + \gamma$$

2. *Primal feasibility*:

$$-p_i^* \leq 0 \quad \forall i$$

$$p_i^* - 1 \leq 0 \quad \forall i$$

$$\sum_i p_i^* = k$$

All primal feasibility conditions are satisfied by our construction of $\mathbf{p}^*$.

3. *Dual feasibility*:

$$\alpha_i \geq 0 \quad \forall i$$

$$\beta_i \geq 0 \quad \forall i$$

$$\gamma \geq 0$$

4. *Complementary slackness*:

$$\alpha_i p_i^* = 0 \quad \forall i$$

$$\beta_i (p_i^* - 1) = 0 \quad \forall i$$

We consider three cases:

- $p_i^* \in (0, 1)$: This means that

$$p_i^* = \frac{\eta^* + \log(w_i u_i)}{|q| u_i},$$

which in turn implies

$$\frac{\partial}{\partial p_i} M_{\text{Kolm}}(\mathbf{u} \odot \mathbf{p}; \mathbf{w}, q) \Big|_{\mathbf{p}=\mathbf{p}^*} = \frac{w_i \exp(-\eta^*)/(w_i u_i)}{\sum_i w_i \exp(q u_i p_i^*)} \cdot u_i$$
$$= \frac{\exp(-\eta^*)}{\sum_i w_i \exp(q u_i p_i^*)}$$

Moreover, due to complementary slackness, we have $\alpha_i = 0$ and $\beta_i = 0$. Thus, for stationarity to hold,

$$\gamma = \frac{\exp(-\eta^*)}{\sum_i w_i \exp(q u_i p_i^*)},$$

which we note is independent of the index $i$.

- $p_i^* = 0$: This implies that

$$p_i^* = 0 \geq \frac{\eta^* + \log(w_i u_i)}{|q| u_i}.$$

Since $\partial M_{\text{Kolm}}/\partial p_i$ is a decreasing function of $p_i$, we have

$$-\frac{\partial}{\partial p_i} M_{\text{Kolm}}(\mathbf{u} \odot \mathbf{p}; \mathbf{w}, q) \Big|_{\mathbf{p}=\mathbf{p}^*} \geq -\frac{\exp(-\eta^*)}{\sum_i w_i \exp(q u_i p_i^*)}$$
$$-\frac{\partial}{\partial p_i} M_{\text{Kolm}}(\mathbf{u} \odot \mathbf{p}; \mathbf{w}, q) \Big|_{\mathbf{p}=\mathbf{p}^*} + \gamma \geq 0$$

Complementary slackness implies that $\beta_i = 0$. For stationarity to hold, we should thus have

$$0 = -\frac{\partial}{\partial p_i} M_{\text{Kolm}}(\mathbf{u} \odot \mathbf{p}; \mathbf{w}, q) - \alpha_i + \gamma,$$

which can be achieved by some $\alpha_i > 0$. Thus, dual feasibility is also satisfied.

- $p_i^* = 1$: This implies that

$$p_i^* = 1 \leq \frac{\eta^* + \log(w_i u_i)}{|q| u_i}.$$

Since $\partial M_{\text{Kolm}}/\partial p_i$ is a decreasing function of $p_i$, we have

$$-\frac{\partial}{\partial p_i} M_{\text{Kolm}}(\mathbf{u} \odot \mathbf{p}; \mathbf{w}, q) \Big|_{\mathbf{p}=\mathbf{p}^*} \leq -\frac{\exp(-\eta^*)}{\sum_i w_i \exp(q u_i p_i^*)}$$
$$-\frac{\partial}{\partial p_i} M_{\text{Kolm}}(\mathbf{u} \odot \mathbf{p}; \mathbf{w}, q) \Big|_{\mathbf{p}=\mathbf{p}^*} + \gamma \leq 0$$

Complementary slackness implies that $\alpha_i = 0$. For stationarity to hold, we should thus have

$$0 = -\frac{\partial}{\partial p_i} M_{\text{Kolm}}(\mathbf{u} \odot \mathbf{p}; \mathbf{w}, q) + \beta_i + \gamma,$$

which can be achieved by some $\beta_i > 0$. Thus, dual feasibility is also satisfied.

Thus, the necessary conditions for KKT are satisfied. Since the constraints on our objective are all affine functions, the necessary conditions are also sufficient. This means that $\mathbf{p}^*$ is the optimal solution.

### C.2.3. ALGORITHM

We describe the algorithm for the Kolm allocation oracle in Algorithm 3. Intuitively, the algorithm proceeds by calculating filling rates for allocation probabilities over the individuals. The critical difference from the WPM algorithm is that since $\lambda$ is an additive factor here, we keep track of two events: 1) when individual $i$'s probability first becomes non-zero, at which point they are included in the active set; and 2) when individual $i$'s probability becomes $p_i = 1$, at which point they are excluded from the active set.

When the fastest-filling individual $i$ reaches $p_i = 1$, that individual is excluded from further allocation. This proceeds until $\sum_i p_i = k$. The sorting of the water-filling rates (Line 17) dominates the time complexity, resulting in an $\mathcal{O}(n \log n)$ total complexity. However, we implement a $\mathcal{O}(n^2)$ algorithm for simplicity here.

---

**Algorithm 3** Kolm Social Welfare Allocation

---

1: **Input:** Utilities $u \in \mathbb{R}^n$, weights $w$, power parameter $q \leq 0$, resources $k$
2: **Output:** Allocation probabilities $p \in [0, 1]^n$
3: **// Step 1: Compute individual allocation rates**
4: **if** $q = -\infty$ **then**
5:     $r_i \leftarrow 1/u_i$ for all $i$
6: **else if** $q = 0$ **then**
7:     $p \leftarrow$ set 1.0 for $k$ indices with largest $(w_i \cdot u_i)$, else 0.0; **return** $p$
8: **else**
9:     $r_i \leftarrow 1/(-q \cdot u_i)$ for all $i$
10: **end if**
11: **// Step 2: Identify critical event times (hitting 0 or 1)**
12: **if** $q = -\infty$ **then**
13:     $T_i^{start} \leftarrow 0, T_i^{end} \leftarrow u_i$ for all $i$
14: **else**
15:     $T_i^{start} \leftarrow -\ln(w_i u_i), T_i^{end} \leftarrow -q u_i - \ln(w_i u_i)$ for all $i$
16: **end if**
17: Sort all $2n$ times $\{T_i^{start}, T_i^{end}\}$ into a sequence $\tau_1, \tau_2, \ldots, \tau_{2n}$
18: **// Step 3: Water-filling over time intervals**
19: Initialize $p \leftarrow \mathbf{0}$, ActiveArms $\leftarrow \emptyset$, $t_{prev} \leftarrow \tau_1$
20: **for** $j = 2$ to $2n$ **do**
21:     $\Delta t \leftarrow \tau_j - t_{prev}$
22:     $m_{interval} \leftarrow \sum_{i \in \text{ActiveArms}} r_i \cdot \Delta t$
23:     **if** $\text{sum}(p) + m_{interval} > k$ **then**
24:         $\Delta t_{final} \leftarrow (k - \text{sum}(p))/\sum_{i \in \text{ActiveArms}} r_i$
25:         $p_i \leftarrow p_i + r_i \cdot \Delta t_{final}$ for all $i \in$ ActiveArms; **break**
26:     **else**
27:         $p_i \leftarrow p_i + r_i \cdot \Delta t$ for all $i \in$ ActiveArms
28:         Update ActiveArms based on whether $\tau_j$ is a $T^{start}$ (add) or $T^{end}$ (remove)
29:         $t_{prev} \leftarrow \tau_j$
30:     **end if**
31: **end for**
32: **return** $p$

---

### C.3. Gini SWF

For the Gini SWF, the optimization objective is:

$$\text{maximize} \quad M_{\text{Gini}}(\boldsymbol{\mu}^\uparrow \odot \mathbf{p}; \mathbf{w}) = \sum_i w_i (\boldsymbol{\mu}^\uparrow \odot \mathbf{p})_{(i)}$$

$$\text{s.t.} \quad p_i \in [0, 1] \quad \forall i$$

$$\sum_i p_i = k$$

Here, the vector of weights $\mathbf{w}$ is such that $w_1 \geq w_2 \geq \ldots \geq w_n \geq 0$. Moreover, $(\boldsymbol{\mu}^\uparrow \odot \mathbf{p})_{(i)}$ specifies the $i$-th order statistic for the vector $\mathbf{u} \odot \mathbf{p}$.

### C.3.1. OPTIMAL ORDER OF INDIVIDUALS

**Proposition C.1.** *There exists an optimal solution $\mathbf{p}^*$ such that, if $\pi$ is the permutation such that*

$$\mu^\uparrow_{\pi(1)} \leq \mu^\uparrow_{\pi(2)} \leq \ldots \leq \mu^\uparrow_{\pi(n)}, \tag{3}$$

*then*

$$\mu^\uparrow_{\pi(1)} p^*_{\pi(1)} \leq \mu^\uparrow_{\pi(2)} p^*_{\pi(2)} \leq \ldots \mu^\uparrow_{\pi(n)} p^*_{\pi(n)}. \tag{4}$$

*That is, the optimal order for $\mu^\uparrow_{\pi(i)} p_{\pi(i)}$ is along a non-decreasing order of $\mu^\uparrow_{\pi(i)}$.*

*Interpretation*: We may index individuals by non-decreasing $\mu_i$ and restrict attention to allocations with $\mu_i p_i$ non-decreasing; then optimizing over $p$ suffices.

*Proof.* For ease of notation, we relabel the indices such that $\pi(i) = i$. Thus, $\mu^\uparrow_1 \leq \mu^\uparrow_2 \leq \ldots \mu^\uparrow_n$.

Let $\mathbf{p}$ be the optimal solution. Let $u_i = \mu_i p_i$, and let $\sigma$ be a permutation such that $\sigma(i)$ is the $i$-th smallest element of $\{u_k\}$. We choose $\sigma$ such that ties are broken to minimize inversions, i.e., $i < j$ such that $u_i > u_j$.

When the $u_i$'s are sorted according to $\sigma$, we get 'blocks' of equal values of $u_i$. Since inversions are minimized, each block has its $\mu_i$'s in non-decreasing order.

If these blocks are such that $\mu_i$'s are in non-decreasing order across the blocks, we are done. Otherwise, let there be two blocks $B_1$ and $B_2$ (with $B_1$ containing smaller values of $u_i$ than $B_2$) such that $\mu_i$'s are not in non-decreasing order across these blocks.

Let $r$ be the rightmost element of $B_1$ and $l$ be the leftmost element of $B_2$. We thus have $u_r < u_l$ (since they belong to distinct blocks) and $\mu_r > \mu_l$ (since $\mu_i$'s are not in non-decreasing order).

Let $r'$ be the element to the right of $r$ when sorted according to $\sigma$, and let $l'$ be the element to the left of $l$. Since $l$ and $r$ form the extremities of their blocks, we must have $u_r < u_{r'}$, and $u_{l'} < u_l$.

Thus, there is some small $\epsilon > 0$ such that $p'_r = p_r + \epsilon \in [0, 1]$, $p'_l = p_l - \epsilon \in [0, 1]$, $u'_r := \mu_r p'_r \leq u_{r'}$, and $u_{l'} \leq u'_l := \mu_l p'_l$. Thus, swapping this small $\epsilon$ probability from $l$ to $r$ does not violate the order constraints. However, this swap results in a change in the Gini SWF of

$$\Delta M_{\text{Gini}} = \epsilon (w_{\sigma(r)} \mu_r - w_{\sigma(l)} \mu_l) \geq 0,$$

since $w_{\sigma(r)} \geq w_{\sigma(l)}$ and $\mu_r > \mu_l$. If $w_{\sigma(r)} > 0$, then this inequality is strict, implying that $\mathbf{p}$ is not optimal. If $w_{\sigma(r)} = 0$, we have $\Delta M_{\text{Gini}} = 0$, which would mean that the swapping of probabilities does not decrease the objective. Repeating this swap (which never decreases the objective) until no such pair of blocks exists yields an optimal solution satisfying $\mu_1 p_1 \leq \ldots \leq \mu_n p_n$. $\qquad \square$

### C.3.2. THEORETICAL PROPERTIES OF ORACLE ALGORITHM

We express the Gini SWF objective as a parametric LP problem. Let $\tau \in [0, k]$ be the parameter controlling the sum of the probabilities. WLOG, we assume that $\mu^\uparrow_i$'s are sorted in non-decreasing order. Our objective is

$$\begin{aligned}
\text{maximize } & \sum_i w_i \mu^\uparrow_i p_i \\
\text{such that } & \sum_i p_i = \tau \\
& p_i \in [0, 1] \quad \forall i \\
& \mu^\uparrow_{i-1} p_{i-1} \leq \mu^\uparrow_i p_i \quad \forall i \in \{2, \ldots, n\}
\end{aligned}$$

Let $\mathbf{p}_\tau$ be the solution encountered by the algorithm for parameter $\tau$. Our algorithm starts at $\tau = 0$, with $\mathbf{p}_0 = \mathbf{0}_n$ being the optimal solution. The algorithm proceeds by choosing a subset of indices $S(\tau)$ for each $\tau$ and increasing the probabilities for these indices as $\tau$ increases, ensuring that no constraints are violated. The set $S(\tau)$ and the rates $r_p(i)$ for the increase in probabilities are chosen to maximize the rate of increase of the Gini objective.

**Proposition C.2.** *The algorithm has the following properties:*

1. *The set of chosen indices and their rates of increase are changed when some element $i$ in $S(\tau)$ has $p_{\tau,i} = 1$.*

2. *The chosen indices for each $\tau$ correspond to a block of consecutive indices $B(\tau)$.*

3. *The choice of indices $B(\tau)$ have their probabilities increased at the rate*

$$r_p(\tau, i) = \frac{1/\mu_i^\uparrow}{\sum_{j \in B(\tau)} 1/\mu_j^\uparrow},$$

   *and this results in a rate of increase of the Gini objective of*

$$r_o(B(\tau)) = \frac{\sum_{i \in B(\tau)} w_i}{\sum_{i \in B(\tau)} 1/\mu_i^\uparrow}$$

4. *Among all feasible infinitesimal directions at any $\tau$, the block $B(\tau)$ chosen by the algorithm and the above rates of increase of the probabilities maximizes the instantaneous rate of increase of the objective.*

5. *At each $\tau$ where the set $B(\tau)$ is changed, we have a sequence of blocks, with the value of $\mu_i p_i$ being the same within a block. These blocks are separated from each other by indices $i$ with $p_i = 1$.*

*Proof.* We prove this result through induction on the set of points where $S(\tau)$ is changed. Since the objective is linear and the constraint set is a convex polytope, this change only occurs a finite number of times.

*Base case*: We begin with $\tau = 0$, where $\mathbf{p}_\tau = \mathbf{0}_n$ is the trivially optimal solution. Part (1) is trivially satisfied here, since this is the first choice of indices and rates we make. Part (5) is also trivially satisfied at $\tau = 0$, since we only have one block. With increasing $\tau$, any permissible increase will involve a suffix block of elements, since the order constraint $\mu_{i-1}p_{i-1} \le \mu_i p_i$ is violated otherwise. Thus, Part (2) is also satisfied.

Let $B$ be some suffix block. Our choice of rate for the probabilities within this block is

$$r_p(i) = \frac{1/\mu_i^\uparrow}{\sum_{j \in B} 1/\mu_j^\uparrow}.$$

This choice results in a rate of increase of utility of

$$r_u(i) = \mu_i^\uparrow \cdot r_p(i) = \frac{1}{\sum_{j \in B} 1/\mu_j^\uparrow}.$$

This rate of increase is the same for all elements within the block, and thus the order constraint is not violated. Moreover, this results in an increase in Gini SWF at the rate

$$r_o(B) = \sum_{i \in B} w_i r_u(i) = \frac{\sum_{i \in B} w_i}{\sum_{i \in B} 1/\mu_i^\uparrow}.$$

Thus, part (3) of our proposition is a consequence of having to satisfy the order constraints.

Let $B'$ be a suffix block which is a subset of $B$ such that $r_o(B') > r_o(B)$, that is, the rate of Gini SWF increase is greater if all the utility is allocated to this sub-block. Then, $B$ is a sub-optimal choice for the block, and we should choose $B'$. Note that switching to a sub-block of $B$ does not violate the order constraint. We can repeat this argument starting from $B = [n]$ to reach a suffix block which has the greatest value of $r_o(B)$. This will be our choice of $B(\tau)$. We have thus characterized this block for Part (4), showing that it provides the maximum rate of increase while satisfying the constraints.

As we increase the probabilities of this block, the leftmost index $l(B)$ is filled fastest, since it has the smallest value of $\mu_i$, and hence the highest filling rate. Thus, this choice of block and filling rates is no longer valid when we have $p_{l(B)} = 1$, which satisfies Part (1). At this instance, we have two blocks of equal values of $\mu_i^{\uparrow} p_i$ (0 for the left block, $\mu_{l(B)}$ for the right) separated by $p_{l(B)} = 1$, which satisfies Part (5).

*Induction case*: Let $\tau$ be some intermediate value where the set of chosen indices and their rates of increase have to be changed. We assume Part (5) holds, i.e., we have a sequence of blocks separated by indices $i$ with $p_i = 1$.

We note that a block in this sequence is similar to the bigger block of $[n]$ for $B = 0$. Thus, if our choice of $S(\tau)$ is to be within a block, all Parts (1)-(5) would be naturally satisfied. We now show that this is indeed the case.

Let us assume the contrary, i.e., our choice of $S(\tau)$ spans across blocks. In this case, the optimal solution will have suffixes from all the blocks. Any feasible infinitesimal direction decomposes as a convex combination of suffix directions within individual blocks. Since the objective slope is linear, an optimal direction concentrates all mass on a single block with maximal slope (ties arbitrary). Thus, Parts (1)-(5) would be satisfied. $\qquad\square$

Because the objective is linear and the feasible region is a convex polytope, following a direction that maximizes instantaneous objective increase keeps the solution on an optimal face of the LP until a constraint becomes tight.

### C.3.3. GINI SWF ORACLE ALGORITHM

We describe the block-based water-filling algorithm in Algorithm 4. The algorithm proceeds by filling probabilities in blocks, ensuring that the order constraints are not violated. The chosen block is such that it provides the greatest rate of increase in the Gini SWF value while following the order constraints. Each block calculation takes $\mathcal{O}(n)$ time, and there are at most $k$ block calculations, resulting in a time complexity of $\mathcal{O}(kn)$.

---

**Algorithm 4** Gini Water-Filling Solver

---

1: **Input:** Vector of mean utilities $u \in \mathbb{R}^n$, weight vector $w$ (where $w_1 \geq w_2 \geq \cdots \geq w_n \geq 0$), number of resources $k$.
2: **Output:** Optimal allocation policy $p \in [0,1]^n$ maximizing $M_{Gini}(\mu \odot p; w)$.
3: Sort $u$ in non-decreasing order: $u_{(1)} \leq u_{(2)} \leq \cdots \leq u_{(n)}$.
4: Let $\pi$ be the permutation such that $u_{(i)} = u_{\pi(i)}$.
5: Initialize $p_i = 0$ for all $i \in [n]$, $rem = k$, and initial block set $\mathcal{B} = \{[1,n]\}$.
6: Precompute suffix sums $W_i = \sum_{j=i}^{n} w_j$ and inverse utility sums $I_i = \sum_{j=i}^{n} 1/u_{(j)}$.
7: **while** $rem > 0$ **do**
8:   Find block $B \in \mathcal{B}$ maximizing the rate: $r(B) = \frac{\sum_{i \in B} w_i}{\sum_{j \in B} 1/u_{(j)}}$.        *Rate of SWF increase*
9:   Let $l(B)$ be the leftmost index of block $B$ and $R$ be the rightmost index.
10:   $\Delta t \leftarrow (1 - p_{\pi(l(B))}) \cdot u_{(l(B))}$        *Time until $p_{l(B)} = 1$*
11:   $m \leftarrow \left( \sum_{j \in B} 1/u_{(j)} \right) \cdot \Delta t$        *Total probability mass required to fill block $B$*
12:   **if** $m \leq rem$ **then**
13:     **for** $j \in B$ **do**
14:       $p_{\pi(j)} \leftarrow p_{\pi(j)} + \Delta t / u_{(j)}$
15:     **end for**
16:     $rem \leftarrow rem - m$
17:     $\mathcal{B} \leftarrow (\mathcal{B} \setminus \{B\}) \cup \{[l(B)+1, R]\}$        *Leftmost index is now fully allocated*
18:     Update precomputed suffix sums for remaining sub-blocks.
19:   **else**
20:     rate_sum $\leftarrow \sum_{j \in B} 1/u_{(j)}$
21:     **for** $j \in B$ **do**
22:       $p_{\pi(j)} \leftarrow p_{\pi(j)} + \frac{rem}{\text{rate\_sum} \cdot u_{(j)}}$
23:     **end for**
24:     $rem \leftarrow 0$
25:   **end if**
26: **end while**
27: **return** $p$

---

**Algorithm 5** $\pi$-ps sampling (pairwise dependent rounding)

**Require:** $\pi \in [0,1]^N$ with $\sum_i \pi_i = n$
**Ensure:** Sample $S \subseteq [N]$, $|S| = n$
 1: **while** there exist at least two fractional entries in $\pi$ **do**
 2:     pick distinct fractional indices $i \neq j$
 3:     $s \leftarrow \pi_i + \pi_j$,    draw $r \sim \text{Unif}[0,1]$
 4:     **if** $s \leq 1$ **then**
 5:       $(\pi_i, \pi_j) \leftarrow \begin{cases} (0, s) & \text{w.p. } \pi_j/s \\ (s, 0) & \text{w.p. } \pi_i/s \end{cases}$
 6:     **else**
 7:       $(\pi_i, \pi_j) \leftarrow \begin{cases} (1, s-1) & \text{w.p. } (1-\pi_j)/(2-s) \\ (s-1, 1) & \text{w.p. } (1-\pi_i)/(2-s) \end{cases}$
 8:     **end if**
 9: **end while**
10:
11: **return** $S \leftarrow \{i : \pi_i = 1\}$

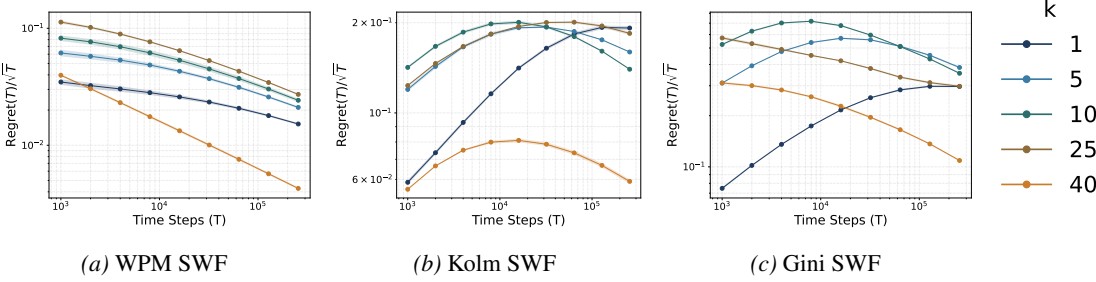

| (a) WPM SWF | (b) Kolm SWF | (c) Gini SWF |

*Figure 4.* Normalized regret $R(T)/\sqrt{T}$ versus time horizon $T$ for WPM, Kolm, and Gini SWFs with linear decay in weights. The normalized regret remains bounded across two orders of magnitude in $T$, consistent with our theoretical $\tilde{\mathcal{O}}(\sqrt{T})$ guarantee.

## D. Algorithm for Dependent Rounding

We use the dependent rounding algorithm (Gandhi et al., 2006; Grafström, 2010) to sample a set $S_t$ given the allocation policy vector $\mathbf{p}_t$ such that $\mathbb{P}(i \in S_t) = p_{t,i}$. This idea is also known as $\pi$-PS sampling, and is popularly used in survey design when samples need to be drawn from a population with unequal probabilities across individuals/groups. Algorithm 5 provides the pseudocode for dependent rounding.

## E. More Experiments: Linear Weight Decay

### E.1. Linear weight decay

We considered an exponential decay of the weight vector in Section 6, which puts greater weight on a few individuals. In this section, we consider a gentler linear decay of the weights, with $w_i = (1 + (i-1)/(n-1))/S$, where $S = \sum_{i=1}^n (1 + (i-1)/(n-1))$. We use the exact same experimental setup as in Section 6, with $n = 50$, $U_i \sim 0.1 + 0.9 X_i$, with each $X_i$ being Beta-distributed with parameters $(\alpha_i, \beta_i)$ identical to those in Section 6.

**Varying horizon $T$.** We let $q = -2$ and vary $T$ the range $[10^3, 1.28 \cdot 10^5]$. We plot the regret against $T$ with varying $k$ in Figure 4. For all three SWFs, we observe that the regret roughly scales as $\mathcal{O}(\sqrt{T})$ when compared with the reference dotted line, consistent with our theoretical guarantees. Generally, we find that the regret increases with across $T$, then decreasing as $k$ is further increased. However, the relative ordering of regret with changing $k$ varies with increasing $T$. We also observe that the trends are quite different from those in Figure 1; In general, we observe a faster decrease in the normalized regret with increasing $T$, indicating that the optimal allocation vector is easier to learn.

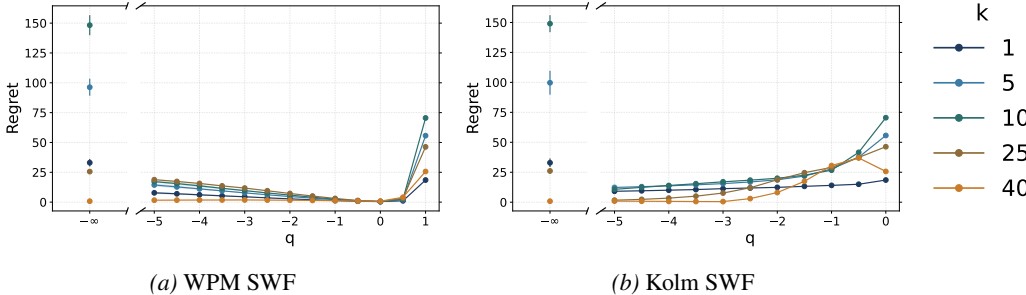

*(a)* WPM SWF          *(b)* Kolm SWF

*Figure 5.* Trends with varying power value $q$ for WPM and Kolm SWFs with linear decay in weights. $q = -\infty$ corresponds to egalitarian welfare for both, whereas $q = 1$ for SWF and $q = 0$ for Kolm correspond to utilitarian welfare. We observe that while there is a smooth change in the observed regret, there is significant variability with changing $k$.

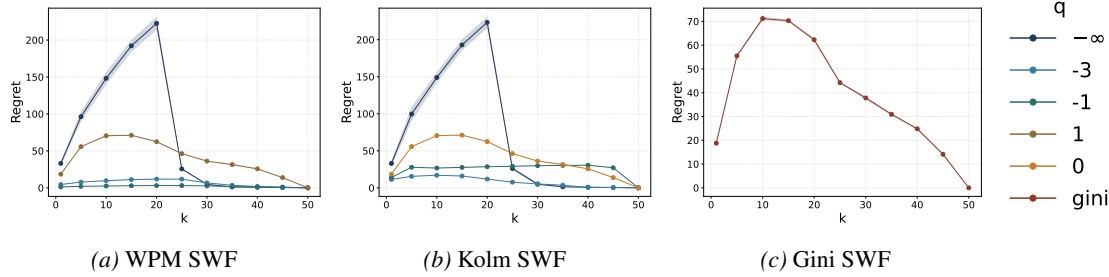

*(a)* WPM SWF         *(b)* Kolm SWF         *(c)* Gini SWF

*Figure 6.* Trends with varying number of allocated resources $k$ for all three SWFs. We observe that there is a sharp decrease after $k = 20$ for egalitarian welfare. The changes with increasing $q$ becomes less gradual for both WPM and Kolm SWFs. Linear weights have some similarity with utilitarian welfare, and we see a similar pattern with varying $k$ for Gini SWF.

**Varying power value $q$.** We now plot the observed regret with different values of $q$ while varying the number of allocated resources $k$ in Figure 5. The regret values are the same for $q = -\infty$ for both WPM and Kolm, which is expected since the weights do not matter for egalitarian welfare. However, the trends for finite $q$ are significantly different. We observe that there is significantly more order in the regret for WPM SWF with increasing $q$. For Kolm SWF, the trends for different $k$ are significantly more unstructured, a phenomenon not seen in the exponential weight decay case.

**Varying number of allocated resources $k$.** We plot the observed regret with different number of allocated resources per time step $k$ in Figure 6. For the WPM SWF, there is a more marked increase in regret with $k$ for finite $q$, which is different from the non-monotonic behavior observed in Figure 3. The Kolm SWF also exhibits similar behavior. The Gini SWF resembles utilitarian welfare, which corresponds to $q = 1$ and $q = 0$ for WPM SWF and Kolm SWF respectively.

### E.2. Runtime measurement

We measure the runtime for SWF-UCB with our implementations of the oracle algorithms for a population of $n = 1000$ individuals. We vary the number of available resources as $k \in \{250, 500, 750\}$. Each experiment is conducted over a horizon of $T = 10,000$ time steps and repeated over five seeded runs. These experiments were conducted on an Apple M2 Pro CPU, and we report the average milliseconds per iteration below:

| **SWF** | $k = 250$ (ms/it) | $k = 500$ (ms/it) | $k = 750$ (ms/it) |
|---|---|---|---|
| WPM | $50.493 \pm 1.727$ | $55.783 \pm 0.943$ | $59.854 \pm 2.636$ |
| Kolm | $56.581 \pm 0.377$ | $70.340 \pm 3.445$ | $82.140 \pm 0.794$ |
| Gini | $51.313 \pm 2.308$ | $53.173 \pm 1.783$ | $69.497 \pm 3.207$ |

*Table 1.* Runtimes for SWF-UCB for all three SWFs with a population of $n = 1000$ individuals over $T = 10,000$ steps.

We note that each iteration includes both the policy optimization (which uses our oracle algorithms) and the allocation

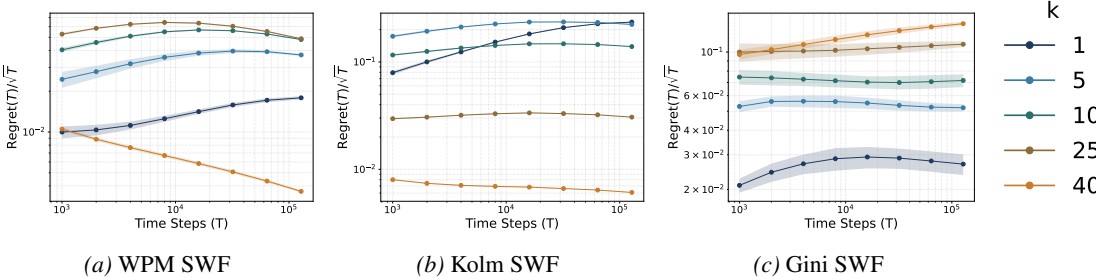

*Figure 7.* Normalized regret $R(T)/\sqrt{T}$ versus time horizon $T$ for WPM, Kolm, and Gini SWFs with heavy-tailed (Pareto-distributed) utilities. The normalized regret remains bounded across two orders of magnitude in $T$, consistent with our theoretical $\tilde{\mathcal{O}}(\sqrt{T})$ guarantee.

sampling (which uses dependent rounding). We observe that each iteration of the algorithm takes less than 85 ms per iteration, indicating that these algorithms are indeed highly efficient.

### E.3. Robustness to heavy-tailed utilities

In Theorem 5.2 we assume that the individual utilities are 1-sub-Gaussian distributed for all $i$. However, this result can be further extended to heavy-tailed utility distributions given valid confidence sequences for these distribtuions. (Wang & Ramdas, 2023) provide bounds on heavy-tailed distributions with known variance bound $\sigma^2$ of the order $\mathcal{O}(\sqrt{\log \log T / T})$. The bounded-variance condition is significantly more general than sub-Gaussianity and includes several heavy-tailed distributions.

We conduct experiments on Pareto distributions with scale set to 0.1 and shape set to $\alpha \in [3, 8]$, resulting in heavy-tailed distributions on $[0.1, \infty)$. We set $n = 50$ and $k \in \{1, 5, 10, 25, 40\}$. We measure the variation in $R(T)/\sqrt{T}$ with the horizon $T$, conducting five seeded run for each experiment. We present our results in Figure 7. We observe that $R(T)/\sqrt{T}$ is bounded, indicating that the regret is $\tilde{\mathcal{O}}(\sqrt{T})$.

## F. Ex-post welfare

As noted in Section 3, we chiefly study ex-ante welfare in this work. Ex-ante welfare is more useful for optimizing per-instance allocation, where we want to optimize the expected welfare prior to allocation.

However, it is also possible to consider *ex-post* notions of welfare. One way to define ex-post welfare is as the welfare over time-averaged realized utilities. It is given by

$$M \left( \frac{1}{T} \sum_{t=1}^{T} \mathbf{u}_t \odot \mathbf{a}_t \right),$$

where $\mathbf{u}_t \sim \mathcal{D}$ are the realized utilities, and $\mathbf{a}_t \sim \mathbf{p}_t$ is the realized allocation vector. Ex-post welfare might be more suitable in time-averaged settings, where we consider the welfare over average realized utilities.

A reasonable baseline is the single best allocation maximizing the limit of post-hoc allocation, and is thus given by

$$\mathbf{p}^* = \arg \max_{\mathbf{p} \sim \mathcal{P}_k} M(\mu \odot \mathbf{p}),$$

where $\mu = \mathbb{E}_{\mathbf{U}_t \sim \mathcal{D}} \mathbf{U}_t$. We measure the ex-post regret as

$$R_{\text{ex-post}}(T) = \mathbb{E}\left[ M\left( \frac{1}{T} \sum_{t=1}^{T} \mathbf{U}_t \odot \mathbf{A}_t^* \right) \right] - \mathbb{E}\left[ M\left( \frac{1}{T} \sum_{t=1}^{T} \mathbf{V}_t \odot \mathbf{A}_t \right) \right],$$

where $\mathbf{U}_t, \mathbf{V}_t \sim \mathcal{D}$, $\mathbf{A}_t^* \sim \mathbf{p}^*$, and $\mathbf{A}_t \sim \mathbf{p}_t$.

This notion of ex-post regret is a harder quantity to analyze. We get the following upper-bound on the ex-post regret

**Theorem F.1.** *Let the utility distributions $\mathcal{D}_i$ be 1-sub-Gaussian for all $i$. Let $M(\mathbf{v})$ be L-Lipschitz continuous w.r.t. the $\ell_\infty$ norm. Let $\Gamma_t$ be the set of all couplings from $\mathbf{A}_t$ to $\mathbf{A}_t^*$ for each $t$. Then,*

$$R_{\text{ex-post}}(T) \leq \frac{L}{T} \sum_{t=1}^{T} \inf_{\Gamma_t} \sum_{i=1}^{n} \mu_i \mathbb{P}(A_{t,i}^* \neq A_{t,i}).$$

*Proof.*

$$\mathbb{E}\left[ M\left( \frac{1}{T} \sum_{t=1}^{T} \mathbf{U}_t \odot \mathbf{A}_t^* \right) \right] - \mathbb{E}\left[ M\left( \frac{1}{T} \sum_{t=1}^{T} \mathbf{V}_t \odot \mathbf{A}_t \right) \right]$$

$$\leq L \sup_{f \text{ 1-Lips}} \mathbb{E}\left[ f\left( \frac{1}{T} \sum_{t=1}^{T} \mathbf{U}_t \odot \mathbf{A}_t^* \right) \right] - \mathbb{E}\left[ f\left( \frac{1}{T} \sum_{t=1}^{T} \mathbf{V}_t \odot \mathbf{A}_t \right) \right]$$

$$\stackrel{(i)}{=} L \inf_{\Gamma} \mathbb{E}\left\| \frac{1}{T} \sum_{t=1}^{T} \mathbf{U}_t \odot \mathbf{A}_t^* - \frac{1}{T} \sum_{t=1}^{T} \mathbf{V}_t \odot \mathbf{A}_t \right\|_1$$

$$\stackrel{(ii)}{\leq} L \inf_{\Gamma} \mathbb{E}\left\| \frac{1}{T} \sum_{t=1}^{T} \mathbf{U}_t \odot \mathbf{A}_t^* - \frac{1}{T} \sum_{t=1}^{T} \mathbf{U}_t \odot \mathbf{A}_t \right\|_1$$

$$\leq \frac{L}{T} \inf_{\Gamma} \sum_{t=1}^{T} \mathbb{E}\left\| \mathbf{U}_t \odot \mathbf{A}_t^* - \mathbf{U}_t \odot \mathbf{A}_t \right\|_1$$

$$\stackrel{(iii)}{\leq} \frac{L}{T} \inf_{\Gamma} \sum_{t=1}^{T} \mathbb{E}\left[ \sum_{i=1}^{n} U_{t,i} \left| A_{t,i}^* - A_{t,i} \right| \right]$$

$$\leq \frac{L}{T} \inf_{\Gamma} \sum_{t=1}^{T} \sum_{i=1}^{n} \mu_i \mathbb{E}\left| A_{t,i}^* - A_{t,i} \right|$$

$$\leq \frac{L}{T} \inf_{\Gamma} \sum_{t=1}^{T} \sum_{i=1}^{n} \mu_i \mathbb{P}(A_{t,i}^* \neq A_{t,i})$$

$$\stackrel{(iv)}{\leq} \frac{L}{T} \sum_{t=1}^{T} \inf_{\Gamma_t} \sum_{i=1}^{n} \mu_i \mathbb{P}(A_{t,i}^* \neq A_{t,i})$$

where we get:

- $(i)$ by the definition of the Wasserstein $W_1$ distance

- $(ii)$ by choosing the potentially sub-optimal coupling resulting in $\mathbf{U}_t = \mathbf{V}_t$

- $(iii)$ from assuming $U_{t,i} \geq 0$

- $(iv)$ by observing that the coupling $\Gamma$ can be decomposed into smaller couplings $\Gamma_t$ which are independent of each other, and taking an infimum over each of them separately

$\square$

This quantity can further be upper-bounded by analyzing the properties of the particular allocation sampling algorithm used (like dependent rounding).

