# OpenReview forum: "Online Social Welfare Function-based Resource Allocation"
_ICML.cc/2026/Conference — ICML 2026 regular_

### Official Review · Reviewer_rPY7 · 2026-02-24

**Soundness:** 4
**Presentation:** 3
**Significance:** 3
**Originality:** 3
**Overall Recommendation:** 5
**Confidence:** 2

**Summary:**

This paper studies repeated allocation of $k \le n$ indivisible resources over $T$ rounds to a fixed population with unknown mean utilities $\mu_i$. In each round, the learner chooses an allocation probability vector $p$ satisfying $\sum_i p_i = k$ and evaluates it through a social welfare function applied to the ex-ante utilities. The main technical contribution is a confidence-sequence lifting theorem showing that coordinate-wise confidence sequences for the $\mu_i$ can be lifted to anytime-valid bounds on the optimal welfare. Building on this result, the paper proposes SWF-UCB, an SWF-agnostic optimistic algorithm with near-optimal regret $\tilde{O}(n+\sqrt{nkT})$, together with exact optimization oracles for the Weighted Power Mean, Kolm, and Gini families. Simulations support the expected $\sqrt{T}$ scaling and reveal a non-monotone dependence on $k$.

**Compliance With Llm Reviewing Policy:**

Affirmed.

**Final Justification:**

I tend to think the paper can be accepted.

**Key Questions For Authors:**

1. Robustness beyond i.i.d./stationarity: how would the lifting theorem and the regret analysis change under (a) non-stationary means $\mu_{t,i}$, (b) contextual utilities, or (c) Markov or restless dynamics? Which components of the framework survive if the coordinate-wise confidence sequences are replaced with alternative ones?
2. Tighter confidence allocation: the paper uses per-coordinate $(1-\delta/n)$ confidence sequences and a union bound. Have you tried SWF-aware allocation of error budgets $\delta_i$, for example based on weights or rank sensitivity under Gini, and does it materially improve regret or stopping behavior?
3. Objective choice and evaluation: can you offer guidance, or an ablation, on when ex-ante welfare is a good proxy in practice and when ex-post criteria would be more suitable? Is there a natural extension of the method to non-degenerate ex-post welfare objectives?

**Limitations:**

yes

**Strengths And Weaknesses:**

**Strengths**

The most interesting part of the paper, in my view, is the confidence-sequence lifting result. It offers a modular statistical bridge between anytime-valid inference and online welfare optimization, and that bridge is conceptually clean. I also liked that the paper stays general at the top level while still instantiating the framework with concrete exact solvers for several important SWF families.
- The lifting theorem is a neat idea: monotonicity alone lets coordinate-wise confidence sequences be converted into time-uniform welfare bounds, which ties together learning and sequential inference in a fairly elegant way.
- Even though the framework is SWF-agnostic, the paper still gives concrete optimization oracles for Weighted Power Mean, Kolm, and Gini, which makes the contribution feel more actionable.
- The regret guarantee has the expected $\sqrt{nkT}$ flavor up to logarithmic factors, and the additive $n$ term does not seem especially problematic.
- The empirical observation that regret is not monotone in $k$ is interesting in its own right and gives the experiments more value than a pure scaling confirmation.
Weaknesses

**Weaknesses**

My reservations are mostly about the modeling assumptions and the breadth of the empirical evidence. Stationary i.i.d. utilities and sub-Gaussian noise make the theory clean, but many applications involve drift, covariates, and interference. I also think the ex-ante welfare objective is principled, yet not obviously aligned with the ex-post fairness or risk-sensitive criteria that some applications would actually care about.
- The stationarity and i.i.d. assumptions limit applicability to contextual or drifting environments. It would help to clarify which parts of the lifting theorem remain valid if one swaps in other confidence sequences or allows modest non-stationarity.
- Optimizing $M(\mu \odot p)$ avoids degeneracy when $k<n$, but it may still diverge from ex-post fairness or risk-sensitive objectives. More discussion or ablation around this modeling choice would strengthen the practical takeaways.
- The analysis appears somewhat conservative because it uses generic Lipschitz constants and a union-bound allocation of confidence across coordinates. A more SWF-aware allocation could conceivably improve the guarantees, but this direction is not developed.
- The experiments are simulation-only and relatively narrow. Additional robustness checks, for example larger $n$, heavier-tailed noise, or semi-synthetic traces, would make the empirical story more convincing.

---

> ### Author Rebuttal · Authors · 2026-03-31
>
> We thank the reviewer for their valuable feedback, positive comments, and insightful questions. We address their concerns below.
>
> **Beyond stationarity and iid assumptions**:
> Our confidence sequence (CS) framework and policy optimization oracle analysis depends only on valid time-uniform CSs for individual utilities and SWF properties. Thus, *any* valid CSs can be used in our framework for online learning and inference applications.
>
> We comment on contextual utilities, where utilities depend on a covariate $x_t$, available at each time step. Consider the linear setting, where utilities are given by $u_i(x_t)=\langle\theta^{\*}_i, x_t\rangle$: one can construct CSs for $\theta^{\*}_i$ (similar to contextual linear bandits) and obtain sub-linear regret. [1] provide time-uniform CS ellipsoids for the linear setting and [2] provide time-uniform bounds on parameters of generalized linear models using online-to-CS conversions.
>
> Time-uniform guarantees for non-stationary settings and restless dynamics is an active area of research: [3] provide time-uniform guarantees for linear bandits with mixing sub-Gaussian noise and [4] construct confidence ellipsoids for restless linear bandits with $\phi$-mixing assumptions. Through our framework, future improvements in time-uniform bounds for these settings can lead to immediate online resource allocation applications.
>
> **Ex-post vs. ex-ante welfare**:
> Ex-ante welfare is more useful for optimizing per-instance allocation, where we want to optimize the expected welfare *prior* to allocation.
>
> Ex-post welfare can be defined as the welfare over time-averaged realized utilities. This notion is given by
> $$M\left(\frac{1}{T}\sum_{t=1}^{T}\mathbf{u}_t\odot\mathbf{a}_t\right),$$
> where $\mathbf{u}_t\sim\mathcal{D}$ are the realized utilities, and $\mathbf{a}_t\sim\mathbf{p}_t$ is the realized allocation vector. Ex-post welfare is more suitable in time-averaged settings, where we consider the welfare over average realized utilities.
>
> A reasonable baseline is the single best allocation maximizing the limit of ex-post welfare, given by
>
> $$\mathbf{p}^{\*}=\arg\max_{\mathbf{p}}M(\boldsymbol{\mu}\odot \mathbf{p}),$$
>
> where $\boldsymbol{\mu}$=$E_{U_t \sim \mathcal{D}}[U_t]$. We measure the ex-post regret as
>
> $$R_{\text{ex-post}}(T)=E\left[M\left(\frac{1}{T}\sum_{t=1}^{T} U_t\odot A_t^{\*}\right)\right]-E\left[M\left(\frac{1}{T}\sum_{t=1}^{T} V_t \odot A_t\right)\right],$$
>
> where $U_t,V_t\sim\mathcal{D}$, $A_t^{\*}\sim\mathbf{p}^{\*}$, and $A_t \sim p_t$.
> We get the following upper bound on the ex-post regret
> $$R_{\text{ex-post}}(T)\leq\frac{L}{T}\inf_{\Gamma}\sum_{t=1}^{T}\sum_{i=1}^{n}\mu_{i}P(A_{t,i}^{\*}\neq A_{t,i}),$$
> where the infimum is over all couplings of $\mathbf{p}$ and $\mathbf{p}^\*$. This quantity can be further bounded by leveraging properties of the sampling algorithm used.
>
> **SWF-aware allocation of union bound**: We agree that a parameter-aware allocation of the $\delta$ union bound to individual CSs might yield a tighter SWF CS. We tried simple heuristics for WPM ($\delta_i\propto w_i$ and $\delta_i\propto w_i^{1/|q|}$), but they did not significantly outperform uniform allocation. Regardless, we believe that better allocations are possible and leave that analysis to future work.
>
> We comment on tighter family-specific bounds in our response to Reviewer nckW.
>
> **Experiments with heavy-tailed noise**:
> [5] provide $O(\sqrt{\log\log T/T})$ bounds on heavy-tailed distributions with known variance bound $\sigma^2$. The bounded-variance condition is significantly more general than sub-Gaussianity and includes several heavy-tailed distributions.
>
> We conduct experiments on Pareto distributions with scale=0.1 and shape=$\alpha \in [3, 8]$, resulting in heavy-tailed distributions on $[0.1,\infty)$. We set $n=50$, $k\in\{1, 5, 10, 25, 40\}$, and use the same SWF parameters as in the main paper. We measure the variation in $R(T)/\sqrt{T}$ with the horizon $T$, conducting five seeded runs for each experiment. We provide a plot of the results here: https://imgur.com/a/GeR22cG. We observe that $R(T)/\sqrt{T}$ is bounded, i.e., the regret is $\tilde{O}(\sqrt{T})$.
>
> ---
> #### References
> 1. Abbasi-Yadkori, Y., Pál, D. and C. Szepesvári. "Online least squares estimation with self-normalized processes: An application to bandit problems." arXiv preprint arXiv:1102.2670 (2011).
> 2. Clerico, E., Flynn, H., and G. Neu. "Confidence sequences for generalized linear models via regret analysis." arXiv preprint arXiv:2504.16555 (2025).
> 3. Abélès, B., Clerico, E., Flynn, H., and G. Neu. "Linear Bandits with Non-iid Noise." arXiv preprint arXiv:2505.20017 (2025).
> 4. Khaleghi, Azadeh. "Restless Linear Bandits." arXiv preprint arXiv:2405.10817 (2024).
> 5. Wang, H. and A. Ramdas. "Catoni-style confidence sequences for heavy-tailed mean estimation." Stochastic Processes and Their Applications 163 (2023): 168-202.

---

> > ### Author Rebuttal · Reviewer_rPY7 · 2026-04-03
> >
> > I appreciate the authors' efforts in providing a thorough rebuttal. I plan to keep my original ratings.

---

### Official Review · Reviewer_W2SF · 2026-03-12

**Soundness:** 3
**Presentation:** 3
**Significance:** 3
**Originality:** 3
**Overall Recommendation:** 4
**Confidence:** 3

**Summary:**

The paper study the problem of repeatedly allocating k identical resources to n individuals, with $k\le n$. The paper propose an algorithm SWF-UCB to maximize social welfare functions that are monotonous convex and lipschitz continuous achieving a regret of $n+\sqrt{nkT}$. They do so by separately constructing and updating confidence intervals on the utility parameters of the indivisduals, and then using an  UCB  approach on the maximization of the social welfare function using the optimistic utility estimate.


Additionally the paper provides efficient offline oracles for social welfare functions: Weighted Power Mean, Kolm, and Gini.

**Compliance With Llm Reviewing Policy:**

Affirmed.

**Final Justification:**

The rebuttal has addressed my concerns regarding originality

**Key Questions For Authors:**

see Strengths And Weaknesses

**Limitations:**

yes

**Strengths And Weaknesses:**

The paper seems well written and easy to understand. I am however not totally convinced on the true novelty of the paper, particularly for what concern the online setting.  In particular, the feedback on $\mu_i$ seems quite powerful, making the construction of the confidence intervals quite straightforward . The UCB approach seems also quite standard. The optimism extended to the function M is interesting but I am not sure how strong of a contribution it is, as it employs monotonicity, which seems to make the result quite immediate.

---

> ### Author Rebuttal · Authors · 2026-03-31
>
> We thank the reviewer for the careful reading and thoughtful feedback. We address their concerns by topic below.
>
> **Feedback model:**
> We would like to clarify that our feedback structure--observing $u_{t,i}$ for each allocated individual $i \in S_t$--is the canonical semi-bandit feedback model from combinatorial bandits [1-3]. This feedback model is well-established, with contributions in combinatorial bandit literature focusing on the structure of the action space, the reward function formulation, and the algorithms needed to handle them.
> Our work follows this paradigm: the novelty lies in (1) the lifting theorem (Thm. 4.1) that connects individual confidence sequences (CSs) to guarantees on the population-level welfare, (2) the efficient oracles for policy optimization, and (3) the unified framework spanning multiple SWF families.
> While we agree that constructing CSs for individual $\mu_i$ is straightforward [4], the lifting theorem provides a powerful way to extend them to a CS for welfare.
>
> **CS lifting theorem (Thm. 4.1):**
> Although the lifting theorem may seem immediate, we believe that this understates the sublety involved. The target of our CS is $M(\boldsymbol{\mu} \odot \mathbf{p}^{\*})$, where $\mathbf{p}^{\*} = \arg\max_p M(\boldsymbol{\mu} \odot \mathbf{p})$ depends on the unknown $\mu$. Thus, the formulation is not just evaluating a monotone function at a confidence bound; it is constructing anytime-valid inference for an optimized quantity.
> Standard UCB analysis does not require such lifting because the comparator (the best fixed arm) does not depend on the unknown parameter in the same structured way.
>
> The simplicity of the lifting theorem enables its applicability to a much broader class of functions than those considered in the paper. Monotonicity is a very natural assumption for SWFs; thus Thm. 4.1 requires a very minimal assumption to construct CSs for welfare.
>
> **Oracle algorithms (Thm. 5.1):**
> We believe our bespoke, efficient oracles constitute a significant algorithmic contribution that may have been underweighted.
> For WPM and Kolm, the closed-form water-filling solutions are non-trivial to derive: we must characterize the structure of optimal solutions, verify KKT conditions carefully (Appendix C.1–C.2), and develop $O(n \log n)$ algorithms to compute the water level parameter $\lambda$.
> For the Gini SWF, the objective is non-differentiable, so standard KKT analysis does not apply. We develop a novel block-based greedy algorithm (Algorithm 4, Appendix C.3) that runs in $O(kn)$ time. Without these oracles, the framework would be computationally impractical; their efficiency makes the approach deployable.
>
> **Breadth of the CS framework:**
> Our framework unifies three axiomatically distinct SWF families (WPM, Kolm, Gini) under a single analysis and provides near-optimal regret matching the $\Omega(\sqrt{nkT})$ lower bound from [3]. The CS construction naturally supports inference applications like sequential hypothesis testing, optimal stopping, and policy evaluation (Appendix A.1). This distinguishes our work from prior approaches that cover only specific objectives [5-7] or provide learning guarantees alone [3].
>
> We will update the paper to make these contributions more prominent and add more nuanced comparisons to the related work discussed above.
>
> ----
>
> #### References
> 1. Cesa-Bianchi, Nicolo, and Gábor Lugosi. "Combinatorial bandits." Journal of Computer and System Sciences 78.5 (2012): 1404-1422.
> 2. Chen, Wei, Yajun Wang, and Yang Yuan. "Combinatorial multi-armed bandit: General framework and applications." International conference on machine learning. PMLR, 2013.
> 3. Kveton, Branislav, et al. "Tight regret bounds for stochastic combinatorial semi-bandits." Artificial Intelligence and Statistics. PMLR, 2015.
> 4. Howard, Steven R., et al. "Time-uniform, nonparametric, nonasymptotic confidence sequences." The Annals of Statistics 49.2 (2021): 1055-1080.
> 5. Barman, Siddharth, et al. "Fairness and welfare quantification for regret in multi-armed bandits." Proceedings of the AAAI Conference on Artificial Intelligence. Vol. 37. No. 6. 2023.
> 6. Sawarni, Ayush, Soumyabrata Pal, and Siddharth Barman. "Nash regret guarantees for linear bandits." Advances in Neural Information Processing Systems 36 (2023): 33288-33318.
> 7. Krishna, Anand, et al. "p-mean regret for stochastic bandits." Proceedings of the AAAI Conference on Artificial Intelligence. Vol. 39. No. 17. 2025.
> 8. Cousins, Cyrus. "An axiomatic theory of provably-fair welfare-centric machine learning." Advances in Neural Information Processing Systems 34 (2021): 16610-16621.

---

> > ### Author Rebuttal · Reviewer_W2SF · 2026-04-04
> >
> > I thank the authors for their detailed answer, my main concerns have been addressed, and I will adjust my score accordingly.

---

### Official Review · Reviewer_nckW · 2026-03-13

**Soundness:** 3
**Presentation:** 4
**Significance:** 3
**Originality:** 3
**Overall Recommendation:** 4
**Confidence:** 2

**Summary:**

This paper studies online resource allocation over a fixed population, where the objective is not standard cumulative reward but an SWF applied to ex-ante utilities. The main contribution is a unified framework that combines online learning and anytime-valid inference: the paper shows that monotonicity alone is enough to lift coordinate-wise confidence sequences for individual utilities into a confidence sequence for the optimal welfare value; it then builds SWF-UCB on top of this idea and gives efficient optimization oracles for WPM, Kolm, and Gini welfare families, together with a near-optimal regret guarantee of order $\widetilde O(L(n+\sqrt{nkT}))$. The experiments are synthetic but support the predicted $\sqrt{T}$ scaling and highlight a non-monotonic dependence of regret on the number of allocated resources $k$.

**Compliance With Llm Reviewing Policy:**

Affirmed.

**Final Justification:**

My questions have been adequately addressed in the rebuttal. However, I still have some concerns, and I remain mildly positive overall on this paper.

**Key Questions For Authors:**

1. The regret bound is near-optimal in a worst-case sense, but the paper also suggests that much better behavior may be possible in structured special cases. Do the authors expect family-specific regret bounds, with explicit dependence on parameters such as $q$ or $w$, to be feasible?

2. Since the paper motivates the framework partly through inference applications such as sequential testing, stopping, and policy evaluation, would the authors consider adding one lightweight main-text illustration of that aspect?

**Limitations:**

Yes.

**Strengths And Weaknesses:**

**Pros**

1. The confidence-sequence lifting result is simple, clean, and easy to understand, while still being broadly useful for connecting utility-level uncertainty to welfare-level guarantees.

2. The paper clearly separates the roles of monotonicity, concavity, and Lipschitz continuity, which makes the technical development feel organized and helps clarify which assumptions are needed for which part of the analysis.

3. In addition to the general framework, the paper provides an explicit SWF-UCB algorithm, exact optimization oracles for several important welfare families, and regret guarantees that are close to optimal in the worst-case sense.

4. Covering WPM, Kolm, and Gini within the same framework is a real strength, since these correspond to meaningfully different welfare/fairness perspectives and broaden the paper’s relevance.



**Cons**

1. The experiments are fully synthetic and mainly validate qualitative trends predicted by the theory. They are helpful, but they do not yet give a strong sense of robustness or practical usefulness in more realistic settings.

2. The oracle complexity results are useful, but because optimization is performed repeatedly, it would be helpful to have a clearer picture of the actual runtime burden in longer-horizon regimes.

---

> ### Author Rebuttal · Authors · 2026-03-31
>
> We thank the reviewer for their valuable feedback and suggestions for additional evaluation. We address their concerns below, including results of additional experiments demonstrating the robustness of SWF-UCB and its algorithmic efficiency.
>
> **Robustness to heavy-tailed distributed utilities**:
> [1] provide bounds on heavy-tailed distributions with known variance bound $\sigma^2$ of the order $\mathcal{O}(\sqrt{\log\log T / T})$. The bounded-variance condition is significantly more general than sub-Gaussianity and includes several heavy-tailed distributions.
>
> We conduct experiments on Pareto distributions with scale set to 0.1 and shape set to $\alpha \in [3, 8]$, resulting in heavy-tailed distributions on $[0.1, \infty)$. We set $n=50$, $k \in \{1, 5, 10, 25, 40\}$, and use the same SWF parameters as in the main paper. We measure the variation in $R(T) / \sqrt{T}$ with the horizon $T$, conducting five seeded runs for each experiment. We observe that $R(T)/\sqrt{T}$ is bounded, indicating that the regret is $\tilde{\mathcal{O}}(\sqrt{T})$. We provide a plot of the results here: https://imgur.com/a/GeR22cG.
>
> **Runtime measurements for SWF-UCB**:
> We measure the runtime for SWF-UCB with our implementations of the oracle algorithms for a population of $n=1000$ individuals (a substantially larger population than those considered in the paper's experiments). We vary the number of available resources as $k \in \{ 250, 500, 750 \}$ and use the same SWF parameters as the main paper's experiments. Each experiment is conducted over a horizon of $T=10,000$ time steps and repeated over five seeded runs on an Apple M2 Pro CPU. We report the average milliseconds per iteration below:
>
> | **SWF** | $k=250$ (ms/it) | $k=500$ (ms/it) | $k=750$ (ms/it) |
> |---|---|---|---|
> | WPM  | $50.493 \pm 1.727$ | $55.783 \pm 0.943$ | $59.854 \pm 2.636$ |
> | Kolm  | $56.581 \pm 0.377$ | $70.340 \pm 3.445$ | $82.140 \pm 0.794$ |
> | Gini  | $51.313 \pm 2.308$ | $53.173 \pm 1.783$ | $69.497 \pm 3.207$ |
>
> Each iteration of the algorithm includes both the policy optimization (using our oracle algorithms) and the allocation sampling (using dependent rounding). Each iteration takes less than 85 ms, indicating that these algorithms are indeed highly efficient.
>
> **Tighter family-specific bounds**:
> The primary focus of this paper is providing a highly general framework applicable to any SWF satisfying some natural assumptions. However, it should be possible to obtain tighter family-specific bounds in two ways:
>
> 1. *Tighter Lipschitz contants*: [2] prove tighter Lipschitz constants for the WPM SWF of the form $\frac{1}{w_{\min}^{1/|p|}}$ for $p < 0$. Tighter family-specific bounds might similarly be obtained for Kolm and Gini SWFs, which would result in better regret bounds.
> 2. *Extension to Hölder continuity*: The proof for Thm. 5.2 (Appendix B.3) can also be modified to account for $\alpha$-Hölder continuity in the $\ell_{\infty}$ norm. We hypothesize that standard techniques can result in bounds of the order $\tilde{\mathcal{O}}(T^{1-\alpha/2})$.
>
> **Main-text illustration for inference applications**:
> We appreciate this suggestion and agree that a main-text illustration would better showcase the inference applications, which were delegated to the appendix due to space constraints. We will add such an illustration in the camera-ready version of the paper if accepted.
>
> ---
> #### References
>
> 1. Wang, Hongjian, and Aaditya Ramdas. "Catoni-style confidence sequences for heavy-tailed mean estimation." Stochastic Processes and Their Applications 163 (2023): 168-202.
> 2. Cousins, Cyrus. "Revisiting fair-PAC learning and the axioms of cardinal welfare." Proceedings of The 26th International Conference on Artificial Intelligence and Statistics, PMLR 206 (2023): 6422-6442.

---

> > ### Author Rebuttal · Reviewer_nckW · 2026-04-04
> >
> > I appreciate the authors' effort in providing a thorough and thoughtful response, and my questions have been adequately addressed. I remain mildly positive on this paper and maintain my current score.

---

### Official Review · Reviewer_pULH · 2026-03-14

**Soundness:** 3
**Presentation:** 4
**Significance:** 3
**Originality:** 3
**Overall Recommendation:** 5
**Confidence:** 2

**Summary:**

This work formalize the classic problem of allocating k resources at each step for T time steps in the lens of online learning. Under this framework, they mainly consider three types of social welfare function, which cover a wide range of popular choices such as utilitarian swf or gini index. These classes of functions satisfy their assumption of monotonicity, concavity, and Lipschitz continuous (with properly chosen parameters). They proposed an online algorithm (framework) inspired by UCB and achieves almost optimal regret (lower bound also provided) $\sqrt{T}$. At the same time, the framework provide uniform confidence intervals that serves the purpose such as sequential hypothesis testing/policy evaluation (whether the current policy achieved desirable performance).

The theoretical results are supported with simulations on synthetic dataset. The simulation results confirm their bound of \sqrt{T} of their algorithm. It also reveals complicated correlation between availability of resources, and the choice of swf.

**Compliance With Llm Reviewing Policy:**

Affirmed.

**Key Questions For Authors:**

Please see strengths and weaknesses.

**Limitations:**

yes

**Strengths And Weaknesses:**

### Strengths:

* The paper is very nicely written and easy to follow. The algorithms and the design of framework are well-motivated and explained in detail.

* The choice of social welfare functions covers a wide range and the result is well connected with sequential hypothesis testing, which provides profound intuitions for practitioners.

* By using the concavity of swf and the lifting technique, the framework is well-motivated and convincing.

* Authors provide both upper and lower bounds for the regret achieved.

### Weakness:

* The paper could benefit from testing on a more realistic setting when utility functions are heterogeneous and have noisy observations, but it is understandable since the main contribution of this paper is proposing a nice framework and theories around it.

* In the paper, authors mentioned the case of "assigning ranger patrols to backcountry zones", it would be interesting to see the comparison with literature studies cases when the population would strategically response to policies such as Hossain, S., Chen, Y., & Chen, Y. (2025). Strategic Hypothesis Testing. arXiv preprint arXiv:2508.03289.

* Here the swf is chosen and fixed, and the fairness concern is implicitly encoded in the objective function. What would be the fairness-efficiency trade-off here? Moreover, since the same objective is applied for several time steps, what can be the possible solution to avoid the ineffectiveness of a fixed measure?

* Writing: use i.i.d. instead of iid, the introduction of functions can be more rigorous by including the support and the range of this function.

---

> ### Author Rebuttal · Authors · 2026-03-31
>
> We thank the reviewer for their thoughtful engagement with our work, their positive assessment of the writing and framework design, and the interesting connection to strategic hypothesis testing. We address each point below.
>
>
> **Heterogeneous utilities and noisy observations:**
> Our framework allows for some heterogeneity in utilities since the utility distributions $\mathcal{D}_i$ can vary across individuals. However, we are happy to discuss further if the reviewer had other forms of heterogeneity in mind.
>
> Moreover, our framework assumes that at each time step $t$, the utility $u_{t,i}$ observed for individual $i$ is randomly distributed according to distribution $\mathcal{D}_i$. This model accounts for (zero-mean) added noise, since it would only change the associated distribution without affecting the mean utility $\mu_i$.  Our experiments draw individual-specific Beta parameters $(\alpha_i, \beta_i)$ independently, yielding distributions with differing means and variances across the population.
>
> We agree that more realistic settings--like contextual utilities or non-stationary distributions--would be valuable extensions. Our confidence sequence (CS) lifting theorem (Thm. 4.1) is agnostic to the specific form of individual-level CSs, suggesting that extensions to contextual bandits or non-stationary settings are natural directions for future work.
>
> **Strategic Hypothesis Testing [1]:**
> In our setting and in [1], a principal decision-maker makes a normative design choice that encodes fairness-efficiency tradeoffs and shapes downstream outcomes. For [1] this is the p-value threshold $\alpha$; for us this is the SWF parameter(s) $q$, $w$.
> However, the strategic channels in both settings differ fundamentally. In [1] agents control the evidence generation process and strategically choose sample sizes given their private beliefs.
> In our setting, we assume that utilities are observed outcomes when resources are allocated and are not strategically constructed by agents. Extending our framework to strategic settings would require modeling different channels, such as strategic utility reporting (mechanism design) or strategic effort in response to allocation (moral hazard). These are substantive extensions that would constitute separate contributions.
> Our framework--the CS lifting theorem (Thm. 4.1) and the efficient oracle algorithms--provides foundational machinery that such extensions could build upon. The lifting theorem is agnostic to the source of uncertainty and requires only valid CSs for individual utilities. This modularity suggests our framework can accommodate richer behavioral models as they are developed.
>
> **Fairness-Efficiency Trade-off:**
> In our setting, this trade-off is directly encoded in the SWF parameters:
>
> - The power parameter $q$ interpolates between utilitarian welfare ($q = 1$ for WPM, $q = 0$ for Kolm), which maximizes aggregate efficiency, and egalitarian welfare $(q = -\infty)$, which prioritizes the worst-off individual.
> - The weight vector $w$ encodes differential importance across individuals, allowing the principal to prioritize vulnerable populations, reflect societal values, or account for group sizes.
>
> Our experiments in Figures 2 and 3 study how learning difficulty (regret) empirically varies across this fairness-efficiency spectrum by varying $q$. The finding that regret behaves non-monotonically in both $q$ and $k$ (Section 6) suggests rich structure in how the fairness-efficiency choice interacts with the online learning problem.
>
> *Effectiveness of a fixed SWF measure over time:*
> The reviewer raises an important point about whether a fixed SWF is appropriate when the same objective is applied over multiple time steps. We see two reasons why the principal decision-maker might change their SWF over time:
>
> 1. They could adaptively select SWF parameters, learning or adjusting $q$ or $w$ over time based on observed outcomes or evolving stakeholder preferences.
> 2. In some applications the principal's normative preferences may genuinely evolve; for instance, shifting from exploration-focused (utilitarian) to equity-focused (egalitarian) objectives as uncertainty decreases.
>
> In both cases, our CS lifting result (Thm. 4.1) and oracle algorithms (Thm. 5.1) would still apply at each time step, though the regret analysis would need to account for a non-stationary objective.
>
> **Writing suggestions:**
> We thank the reviewer for their suggestions to improve mathematical precision. We will replace "iid" with "i.i.d." throughout and add explicit domain and range specifications for all function definitions.
>
> ----
>
> #### References
>
> 1. Hossain, Safwan, Chen, Yatong, & Yiling Chen. (2025). Strategic Hypothesis Testing. arXiv preprint arXiv:2508.03289.

---

> > ### Author Rebuttal · Reviewer_pULH · 2026-04-04
> >
> > I thank the authors for their thoughtful reply and hence keep my positive score.

---

### Decision · Program_Chairs · 2026-04-30

**Decision:**

Accept (regular)

**Comment:**

All the reviewers appreciated the contribution of the paper and the rebuttal has addressed the concerns raised. Personally, I like the idea of the confidence-sequence lifting result and, although similar to classical concentration arguments, it seems to offer a new perspective that can be more generally useful. I also agree with the reviewers about the nice presentation of the paper. I recommend that the paper is accepted.

One feedback from my reading with respect to the choice of ex-ante metrics: I understand that having (within-rounds) ex-post metrics may not make sense (for example, then the minimum across individuals would be 0). That said, we could take the time-average ex-post reward and apply the social-welfare function on those time-averages (arguably, the same individual cares about their cumulative performance). It would be nice if the paper commented on whether such objectives are simpler or pose similar technical challenges. (Even for those objectives, it is not clear to me whether canonical approaches would solve them and it would be interesting to see whether the lifting theorem provided could extend there as well.)